# Automating Environments For Measuring Agentic Learning

## Abstract

Humans naturally adapt to diverse environments by learning underlying rules across worlds with different dynamics, observations, and reward structures. In contrast, existing agents typically demonstrate improvements via self-evolving within a single domain, implicitly assuming a fixed environment distribution. Cross-environment learning has remained largely unmeasured: there is no standard collection of controllable, heterogeneous environments, nor a unified way to represent how agents learn. We address these gaps in two steps. First, we propose AUTOENV, an automated framework that treats environments as factorizable distributions over transitions, observations, and rewards, enabling low-cost ($4.12 on average) generation of heterogeneous worlds. Using AUTOENV, we construct AUTOENV-36, a dataset of 36 environments with 358 validated levels, on which seven language models achieve 12-49% performance, demonstrating the challenge of AUTOENV-36. Second, we formalize agent learning as a component-centric process driven by three stages of Selection, Optimization, and Evaluation applied to an improvable agent component. Using this formulation, we design eight learning methods and evaluate them on AUTOENV-36. Empirically, the gain of any single learning method quickly decreases as the number of environments increases, revealing that fixed learning methods do not scale across heterogeneous environments. Environment-adaptive selection of learning methods improves performance but exhibits diminishing returns as the method space expands. These results highlight both the necessity and the current limitations of agent learning for scalable cross-environment generalization, and position AUTOENV and AUTOENV-36 as a testbed for studying cross-environment agent learning.

## 1 Introduction

The pursuit of intelligent agents that can naturally traverse environments like humans has been a longstanding goal (Sutton & Barto, 1998). Humans seamlessly transition from board games to video games with artificial physics, or from real-world tasks to virtual environments with entirely different rule systems (Liu et al., 2025; Ying et al., 2025). This adaptability stems from humans' natural access to diverse environments and their inherent capacity for learning across environments.

Yet current artificial agents fall short of this ideal. Recent language agents achieve strong performance in individual environments (Liu et al., 2025; Yu et al., 2025), but these abilities mostly come from human-designed training data, reward signals, and agent architectures rather than from learning across environments. Recent work attempts to give agents agentic learning or self-evolving abilities, including prompt optimization (Xiang et al., 2025), agent code optimization (Zhang et al., 2024), and agentic reinforcement learning (Wang et al., 2025b). These methods report strong gains, but almost always within a single environment family, such as coding (Zhang et al., 2025; Jimenez et al., 2024a), search (Team, 2025; Wei et al., 2025), or games (Wang et al., 2023). In contrast, human learning is naturally cross-environment: we place different emphases in our learning when facing environments with different rules over dynamics, observations, and rewards. Thus, the central open question for agent learning is not only whether agents can improve within a single domain, but whether they can learn effectively across heterogeneous environments with different rule distributions, achieving robust cross-environment agent learning. Figure 1 illustrates this contrast.

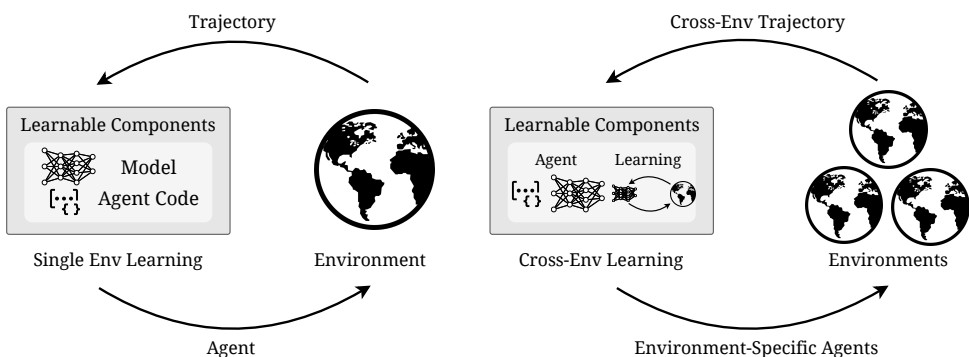

Figure 1: Conceptual comparison between learning in a single environment (left) and cross-environment learning (right). In the single-environment case, trajectories from one environment are used to update the agent's internal components (model and agent code). In the cross-environment case, trajectories collected across many environments update both the agent components and a shared learning procedure, so that the agent learns how to learn from diverse environments.

This question remains largely unexplored because current agent infrastructure has two key gaps. First, agents do not have access to an extensible collection of environments with heterogeneous rule distributions over rewards, observations, and transitions. Most work relies on a small set of human-designed environments, which are hard to scale and cover only a narrow slice of possible rules. Recent efforts in automated environment construction (Shi et al., 2025; Hu et al., 2025b; Fang et al., 2025) mainly synthesize data for specific real-world tasks, aiming to improve a single application rather than to span diverse rule systems for cross-environment generalization. Second, there is no unified way to represent how agents learn. Existing works modify prompts, code, or workflows (Zhang et al., 2025; 2024; Xiang et al., 2025), but these learning methods are encoded as scripts or system prompts, making them hard to compare or reuse across settings. As a result, we lack both diverse environments and a common representation for learning methods which prevents systematic measurement of cross-environment agent learning.

To address the lack of diverse environments, we propose AUTOENV, an automated framework that creates many environments with different rule distributions. AUTOENV treats an environment as a distribution over rewards, states, transitions, and observations, and uses three abstraction layers plus coding agents to implement and validate each level. From its outputs, we create 36 high-quality environments to form AUTOENV-36, with 358 validated levels covering navigation, manipulation, pattern reasoning, and simulation. Environment evaluation shows that seven strong language models reach only 12–49% normalized reward on AUTOENV-36, indicating that it is both challenging and discriminative. Environment generation experiments further show that AUTOENV attains a 90% execution success rate at an average cost of $4.12 per environment.

To explore learning strategies in a structured way, we formalize agentic learning as a component-centric process with three stages: Selection, Optimization, and Evaluation applied to an improvable agent component. In this view, a learning method is a discrete combination of a selection method (Best or Pareto), an optimization signal (from environment dynamics or agent instruction), and a target component (prompts, agent code, or process tools). This definition gives a clear configuration space of learning methods and allows the same learning patterns to be instantiated and compared across different environments.

Empirically, our learning experiments reveal two clear patterns. First, the benefit of any fixed learning method quickly shrinks as environment diversity grows: methods that improve performance by about 8 points on a 6-environment subset yield only around 3 points of gain when applied uniformly across all 36 environments in AUTOENV-36. Second, selecting different learning methods for different environments substantially improves performance but shows diminishing returns as we add more methods. Together, these trends indicate that fixed learning methods do not scale across heterogeneous environments, and that current adaptive schemes are only a first step toward robust cross-environment agent learning.

Our main contributions are threefold. (1) We introduce AUTOENV, an automated environment generation framework with low cost (on average $4.12 per environment), and AUTOENV-36, a dataset of 36 heterogeneous environments with 358 validated levels. (2) We formalize agentic learning as a component-centric framework (Selection, Optimization, and Evaluation) and instantiate eight concrete learning methods within this space. (3) We systematically study how environment diversity affects agent learning, showing that fixed learning methods break down as the number of environments grows, while simple environment-adaptive selection improves cross-environment performance but still leaves a clear gap to the achievable learning-method upper bound.

## 2 RELATED WORK

**Agentic Environment.** Effective agent environments today are largely built by substantial human effort and typically follow a narrow distribution of rules, such as coding Jain et al. (2024); Jimenez et al. (2024b), search Deng et al. (2025); Wei et al. (2025); Mialon et al. (2023), games Fan et al. (2022), and embodied settings Tao et al. (2024); Shridhar et al. (2021). Recent work instead tries to automatically construct agentic environments. One direction keeps the underlying application distribution fixed and scales data within that distribution: systems such as AutoBencher Li et al. (2024), TaskCraft Shi et al. (2025), GG-Bench Verma et al. (2025), and ARE Andrews et al. (2025) automatically generate new tasks, games, or scenarios on top of pre-defined tools or apps while leaving the core rules unchanged. Another direction uses strong models as simulators of existing environments, distilling environment dynamics into world or experience models so that agents can be trained via cheap simulated rollouts instead of interacting with the original system Li et al. (2025); Chen et al. (2025). In contrast, AUTOENV focuses on scaling heterogeneous environments by automatically constructing the environments themselves under diverse rule distributions.

**Agentic Learning.** Recent advances in agentic learning focus on improving prompts, agent code, and underlying models. Prompt-centric methods such as SPO (Xiang et al., 2025), GEPA (Agrawal et al., 2025), and DSPy (Khattab et al., 2023) treat prompts or textual modules as the primary object of optimization, using model- or metric-based feedback (with or without ground-truth labels) to iteratively refine natural-language policies. Agent-code methods such as AFlow (Zhang et al., 2024), Darwin Gödel Machine (Zhang et al., 2025), and Huxley–Gödel Machine (Wang et al., 2025a) view the agent as a mutable program, repeatedly rewriting workflows or coding agents and validating changes on downstream benchmarks. At the model level, methods such as RAGEN (Wang et al., 2025b) and Learn-by-Interact (Su et al., 2025) perform trajectory-level reinforcement learning or data-centric adaptation, updating the underlying policy from interaction traces in realistic environments. However, these works typically fix a single learning strategy within limited environment families, whereas we instead provide a unified formulation of agentic learning as composable selection, signal, and component updates, and use AUTOENV's heterogeneous environments to systematically measure how different learning strategies perform across environments.

## 3 AUTOMATED ENVIRONMENT GENERATION

### 3.1 ENVIRONMENT FORMULATION

To enable systematic generation of diverse environments with distinct rule distributions, we formalize an environment following the definition in (Sutton & Barto, 1998) as a tuple $\mathcal{E} = (\mathcal{S}, \mathcal{A}, T, R, \Omega, \tau)$ where $\mathcal{S}$, $\mathcal{A}$, $T$, $R$, $\Omega$, and $\tau$ represent state space, action space, transition function, reward function, observation function, and termination predicate respectively. Unlike existing symbolic planning formalisms such as PDDL (Hu et al., 2025a), this reinforcement learning-based formulation can be naturally implemented in code and more easily supports diverse environment types including those with accumulative rewards and continuous numerical states.

AUTOENV further decomposes this formulation into three abstraction layers to support systematic environment generation and agent interaction. **BaseEnv** implements the core dynamics $(\mathcal{S}, \mathcal{A}, T, R, \tau)$, capturing the fundamental rules of the environment and maintaining the full world state. On top of this, **ObsEnv** specializes the observation function $\Omega$ through configurable observation policies, given the underlying state and transitions from **BaseEnv**, it decides what information is actually exposed to the agent. This allows us to vary information availability in a controlled way,

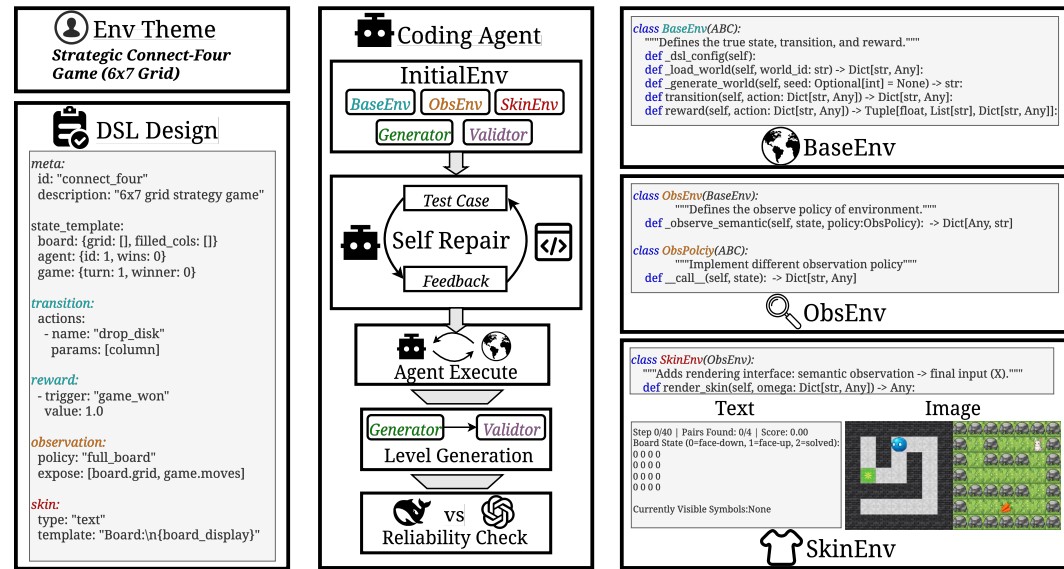

Figure 2: Overview of the AUTOENV environment generation pipeline. The left panel shows an input environment theme and its DSL design in YAML form. The middle panel illustrates how coding agents instantiate basic code from the DSL, then run a self-repair loop followed by three verification stages, execution testing, level generation, and reliability checking with differential models. The right panel presents the core code structure for the three abstraction layers, and an example SkinEnv that renders both text descriptions and an image view.

from full observability to strong partial observability. Finally, **SkinEnv** applies rendering on top of **ObsEnv** and converts observations into agent-facing modalities, such as natural language text or images. The same observation policy can therefore be paired with different skins, producing environments that look very different to the agent while sharing the same underlying rules. This layered design lets us change rule distributions at the dynamics or observation level, or create semantically different views of the same rules, all while keeping a consistent programming interface. The code implementation of these three layers is provided in Appendix A.

## 3.2 ENVIRONMENT GENERATION

**Generation.** AUTOENV automates environment creation with a pipeline that follows the three-layer abstraction. Given an environment theme, AUTOENV first translates the theme into a detailed language description of the environment, including its goals, rules, state variables, and reward conditions. We then convert this description into a YAML file using a domain-specific language (DSL) that specifies the core dynamics for BaseEnv, the observation policy for ObsEnv, the rendering rules for SkinEnv, and the configuration of a level generator. Next, coding agents (Yang et al., 2024) read the DSL and implement three-layer classes, the level generator, the validator, and concise agent-facing documentation (environment instructions and action-space descriptions). After the initial code is produced, we employ the same coding agents in a simple self-repair loop: they run syntax and execution tests on the environment, collect error messages, and iteratively edit the code until the tests pass or a repair budget is reached. The final outputs of this generation stage are executable environment code for all three layers, a level generator that can produce candidate levels, and a validator that includes a *max_reward* function used later for normalized evaluation.

**Verification.** AUTOENV uses a three-stage verification pipeline that matches the three metrics: *Execution, Level Generation,* and *Reliability*. In the execution stage, we run a simple ReAct-style agent in each generated environment for a short rollout to detect compilation and runtime errors. Environments that crash or hang are removed. In the level generation stage, we use the level generator to create multiple candidate levels for each remaining environment and apply environment-specific validators to check basic properties such as goal reachability and reward structure. Environments that can reliably produce valid levels pass this stage. In the reliability stage, we perform differential model testing with two ReAct agents backed by GPT-4o-mini and DeepSeek-V3.1. If the weaker

model consistently achieves higher rewards than the stronger one, we treat the reward structure as unreliable (close to random) and discard that environment. Together, these three stages ensure that the final environments are executable, can generate valid levels, and provide skill-based (non-random) rewards. The resulting file structure produced by this pipeline is detailed in Appendix B.3, and the complete generation process is shown in Figure 2 and detailed in Appendix B.1.

### 3.3 AUTOENV-36 DATASET

Using AUTOENV, we first instantiated environments from 100 different themes with text-only **SkinEnv** rendering. After the full three-stage verification pipeline, 65 environments remained. From this verified pool, we selected 36 environments to form AUTOENV-36 (AUTOENV-36). The selection was based on their rule types and difficulty, we kept environments that covered diverse reward, observation, and semantic patterns, and filtered out environments that were either too trivial or too hard to provide meaningful evaluation.

We organize AUTOENV-36 along three key dimensions. For **Reward**, we distinguish between binary rewards (success/failure at the end) and accumulative rewards (scores collected over time). For **Observation**, we distinguish full observability, where the agent can see all relevant state information, from partial observability, where the agent only sees a subset of the state or local views. For **Semantic** representation, we consider whether the environment description is aligned or inverse. In aligned environments, the natural language semantics match the underlying rules (e.g., "poison decreases health" and "water restores health"). In inverse environments, the semantics are intentionally counterintuitive (e.g., "poison restores health" while "water decreases health"), forcing agents to rely on interaction rather than surface wording.

Table 1: AUTOENV-36 dataset statistics across key environmental dimensions.

| Metrics | Reward | | Observation | | Semantic | | AUTOENV-36 |
|---|---|---|---|---|---|---|---|
| | Binary | Accum. | Full | Partial | Aligned | Inverse | |
| Count | 18 | 18 | 15 | 21 | 28 | 8 | 36 |
| Ratio | 50.0% | 50.0% | 41.7% | 58.3% | 78.8% | 22.2% | 100.0% |
| Actions | 5.33 | 6.78 | 5.47 | 6.48 | 6.14 | 5.75 | 6.10 |
| Code Lines | 407.43 | 504.58 | 456.78 | 449.88 | 469.28 | 414.00 | 471.14 |

As shown in Table 1, AUTOENV-36 achieves balanced coverage across these dimensions. The dataset splits evenly between binary and accumulative rewards (50% each), and slightly favors partial observability (58.3%) to reflect realistic information constraints. Most environments use aligned semantics (78.8%), while a smaller subset (22.2%) uses inverse semantics to test robustness to misleading descriptions. The environments have moderate structural complexity, with an average of 6.10 available actions and 471.14 lines of implementation code. We also provide per-environment feature summaries and the main agent skills each environment targets in Appendix A7. Beyond the text-only settings used in our main experiments, we also explore multi-modal **SkinEnv** generation by combining coding agents with image generation models (Google, 2025). Examples of these multi-modal environments are shown in Appendix A1.

## 4 LEARNING

### 4.1 LEARNING FORMULATION

We view agent learning as a component-centric process, the agent improves by repeatedly updating internal components (prompts, agent code, tools, and model) based on feedback from environment interactions. To make this process explicit and comparable across methods, we formalize it using four basic objects and three stages.

**Basic objects.** A *candidate* $c \in \mathcal{C}$ represents one version of the agent during learning. It contains the current values of its internal components and also stores metadata, including recent trajectories and metrics. A *component* is any part of a candidate that can be modified by learning. By specifying the component type in a learning method, we can update different levels of the agent, for example

only changing the prompt, editing the global agent code, or modifying memory, tool sets, or the underlying model. When a candidate interacts with an environment, it produces a *trajectory* $\tau \in \mathcal{T}$, which records the agent's behavior and feedback from the environment. From this trajectory we compute one or more *metrics* $m \in \mathcal{M}$, such as success rate, number of steps, or token usage.

**Three stages of agent learning.** We describe agent learning as an iterative process with three core stages *Selection*, *Optimization*, and *Evaluation*, and algorithm in Appendix D.2.

- **Selection** chooses which candidates from the current pool should be considered in the next learning step. The selection function $\mathcal{F}_s$ can use different rules, for example picking the best candidates according to their current metrics, using Pareto selection over multiple metrics, or sampling candidates at random to keep diversity.

- **Optimization** takes the selected candidates and produces updated candidates by modifying their target components. The optimization function $\mathcal{F}_o$ uses an optimization model (such as a language model) that can look at the structure of a candidate, its past trajectories, and its metrics, then propose edits to the chosen components. For example, it can rewrite prompts based on scores, edit agent code after inspecting failure cases, or adjust process operators after analyzing patterns in the trajectories.

- **Evaluation** runs candidates in the environment and measures how well they perform. The evaluation function $\mathcal{F}_e$ uses an execution model to run a candidate in environment $\mathcal{E}$ and obtain a trajectory, then applies an evaluation rule to compute metrics. The evaluation rule can be a simple average reward, an LLM-as-a-judge that scores the trajectory, or a trained reward model.

Using these four basic objects and three stages, we can express many existing agent learning methods such as SPO(Xiang et al., 2025) and GEPA(Agrawal et al., 2025) for prompt optimization, AFlow (Zhang et al., 2024) for agentic workflow optimization, and the Darwin Gödel Machine (Zhang et al., 2025) for code-level self-modification. Their detailed formulations in our framework are given in Appendix D.3.

## 4.2 Learning Implementation

Our component-centric formulation is used to define a search space over learning methods. Each learning method is a combination of three design choices, the selection strategy $\mathcal{F}_s$, the optimization scheme $\mathcal{F}_o$, and the target component that can be changed. In this view, agent learning itself becomes something we can search over and compare across environments.

In our experiments, we instantiate eight learning methods by combining two selection rules, two optimization styles, and two target components. For evaluation, we use each environment's native reward and report normalized reward as the main metric. We describe the detail below, and provide the full optimization prompts in Appendix D.4. We also define a *Learning Upper Bound* for each environment, which is the best performance achievable by any method in this eight-method space when we are allowed to choose a different method for each environment. This upper bound acts as an ideal environment-specific learner and lets us measure the gap between any single fixed learning method and the best achievable cross-environment learning behavior under our current method space.

- **Selection functions** $\mathcal{F}_s$. Best Selection keeps the candidate with the highest normalized reward. Pareto Selection keeps all candidates that are not dominated in the space of reward and cost.

- **Optimization schemes** $\mathcal{F}_o$. The dynamics-based scheme analyzes trajectories to extract environment dynamics and failure patterns, then turns them into rules or guidelines that are used to improve the agent. The instruction-based scheme analyzes trajectories to find incorrect behaviors and then rewrites the prompt so that the agent reasons and acts better in the same environment.

- **Evaluation** $\mathcal{F}_e$. We run candidates on multiple levels of an environment using an execution model, collect trajectories, and compute normalized reward using the environment's reward function.

- **Target components**. We support prompt components that encode instructions and agent code components that implement the agent logic, so learning methods can change either how the agent is instructed or how the program behaves.

# 5 EXPERIMENTS

## 5.1 EXPERIMENTAL SETUP

**Models.** For environment generation, we evaluate AUTOENV's generation effectiveness using Claude-4-Sonnet (Anthropic, 2025) as the generation model. For environment evaluation, we test generated environments with GPT-4o-mini (OpenAI, 2024), GPT-5 (OpenAI, 2025a), O3 (OpenAI, 2025b), Claude-4-Sonnet, Kimi-K2 (Kimi, 2025), DeepSeek-V3.1 (DeepSeek, 2025), and Gemini-2.5-Flash (Comanici et al., 2025). For learning experiments, we use Claude-4-Sonnet as the optimization model, use DeepSeek-V3.1, Gemini-2.5-Flash, Qwen-2.5-7B (Qwen et al., 2025) as execution model across experiments.

**Metrics.** For environment generation, we report success rates across generation phases including code execution, level generation, and performance reliability, along with overall success rates and generation costs. For environment evaluation, we report normalized accuracy, standard deviation, and average steps per level. Normalized accuracy is computed as the achieved reward divided by a validator-estimated upper bound on the level reward; because this bound is approximate, agents can occasionally discover strategies that exceed it and thus obtain accuracy values above 100% (see Appendix C.2). All accuracy results are averaged over three runs. For partial learning experiments, we report execution cost ($) to show how selection methods affect cost.

**Implementation Details.** All agents begin with the ReAcT(Yao et al., 2023) framework (IO) with system prompts provided by each environment's agent instruction, with the agent details can be found in Appendix C.3. For learning experiments, we construct training and testing splits for each environment where testing samples are fixed but training samples are dynamically generated using each environment's built-in generator and validator. During each learning iteration's evaluation phase, we execute three runs per sample to ensure stable evaluation signals. We set the maximum learning iterations to 10 rounds.

## 5.2 ANALYSIS ON AUTOENV

**Environment Generation Analysis.** We evaluate AUTOENV's performance using 100 environment themes, where 75 are generated purely by LLM and 25 involve human review and modification. As shown in Table 2, AUTOENV achieves 90.0% execution success, 96.7% level generation success, and 74.7% reliability verification rate through differential model testing, resulting in an overall success rate of 65.0%. Human review of LLM-generated themes significantly improves overall success rates from 60.0% to 80.0%, primarily by reducing rule inconsistencies caused by overly abstract or redundant theme descriptions (see Appendix B.7). Compared to manual environment construction, AUTOENV generates diverse environments at approximately $4.12 per environment, enabling scalable agent environment generation.

Table 2: Evaluation of AUTOENV generation pipeline across 100 environment themes. **Automated**: 75 purely LLM-generated themes; **Supervised**: 25 themes with human review; **Overall**: all themes combined. Overall success rate represents environments passing all validation stages. Cost is measured in USD per environment.

| Source | Success Rate | | | | Cost |
|---|---|---|---|---|---|
| | Execution | Level Generation | Reliability | Overall | |
| Automated | 88.0 | 97.0 | 70.3 | 60.0 | 4.33 |
| Supervised | 96.0 | 95.8 | 87.0 | 80.0 | 3.48 |
| Overall | 90.0 | 96.7 | 74.7 | 65.0 | 4.12 |

**Environment Evaluation Analysis.** As shown in Table 3, AUTOENV-36 demonstrates clear performance differentiation across model capabilities. O3 achieves the highest performance at 48.73%,

Table 3: Model performance on AUTOENV-36. We evaluate 7 language models across 36 environments. Acc: normalized reward, Std: standard deviation over 3 runs, Steps: average action rounds.

| Method | Reward | | Observation | | Semantic | | Avg. | | |
|---|---|---|---|---|---|---|---|---|---|
| | Binary | Accum. | Full | Partial | Aligned | Inverse | Acc | Std | Steps |
| GPT-4o-mini | 7.96 | 15.97 | 12.65 | 11.48 | 12.36 | 10.58 | 11.96 | (±0.37) | 29.79 |
| GPT-5 | 56.25 | **37.35** | 49.81 | 44.65 | 45.44 | 51.55 | 46.80 | (±0.57) | 24.98 |
| o3 | **61.16** | 36.31 | **53.42** | **45.38** | **47.56** | **52.84** | **48.73** | (±0.80) | 24.61 |
| Claude-4-Sonnet | 45.09 | 36.25 | 45.69 | 37.09 | 38.14 | 49.53 | 40.67 | (±0.89) | 24.88 |
| Kimi-K2 | 31.44 | 31.55 | 34.59 | 29.28 | 28.86 | 40.70 | 31.49 | (±1.91) | 26.07 |
| DeepSeek-V3.1 | 34.63 | 33.38 | 35.66 | 32.82 | 32.68 | 38.63 | 34.01 | (±1.42) | 26.93 |
| Gemini-2.5-Flash | 43.89 | 34.93 | 46.84 | 34.10 | 38.95 | 41.02 | 39.41 | (±0.42) | 24.32 |
| Average | 40.06 | 32.25 | 39.81 | 33.54 | 34.86 | 40.69 | 36.15 | – | 25.94 |

followed by GPT-5 at 46.80%, while GPT-4o-mini get the lowest performance at 11.96%. The substantial performance gap across model tiers validates AUTOENV-36 as an effective benchmark for measuring agent capabilities. Analysis across environment characteristics reveals three key patterns. First, binary reward environments (avg. 40.06% across models) consistently outperform accumulative reward environments (avg. 32.25%), likely due to simpler reward triggering mechanisms. Second, full observation environments (avg. 39.81%) outperform partial observation environments (avg. 33.54%), confirming that information availability significantly impacts performance.

Interestingly, inverse semantic environments (avg. 40.69%) yield higher scores than aligned semantic environments (avg. 36.15%), which goes against what we expected. To understand whether this happens because inverse environments are easier, or because of how semantics are shown, we run a controlled experiment in Appendix E.3. We take aligned environments and inverse only their semantic display while keeping the rules the same. Results show that this inverse causes an 68.8% performance drop. This means inverse semantics do make tasks harder, so the higher scores in inverse environments likely happen because those environments were built simpler during generation.

## 5.3 ANALYSIS ON AGENT LEARNING

We conduct experiments to analyze how environment diversity and learning method diversity affect agent learning. We evaluate on two scales: (1) a 6-environment subset (selected with feature diversity, detailed in Appendix D.1) for learning method analysis, and (2) the full 36 environments for environment diversity analysis.

Table 4: Learning method analysis on 6 environments using Qwen-2.5-7B. We test 5 methods including 4 training-free approaches and SFT. Upper bound selects the best method per environment. Full environment names in Appendix A6, and the SFT detailed in Appendix E.2.

| Method | Learning Subset | | | | | | Avg. |
|---|---|---|---|---|---|---|---|
| | 1-ID | 19-AS | 21-WN | 24-MM | 26-TA | 33-AD | |
| *IO* | | | | | | | |
| Qwen-2.5-7B | 12.23 | 20.16 | 3.33 | 0.00 | 35.08 | 34.39 | 17.53 |
| **Learning** | | | | | | | |
| Dynamics + PO | 15.22 | 20.22 | **6.67** | 0.00 | 35.08 | 36.76 | 18.89 |
| Instruction + PO | 29.76 | 20.76 | 3.33 | 0.00 | 32.77 | 32.52 | 19.86 |
| Dynamics + Agent | **31.37** | 20.44 | 3.33 | 0.00 | 31.74 | 61.61 | 24.75 |
| Instruction + Agent | 12.23 | 20.18 | 3.33 | 0.00 | **49.02** | 49.21 | 22.33 |
| SFT | 25.93 | **24.34** | 3.33 | 0.00 | 35.15 | **61.78** | **25.09** |
| UpperBound | 31.37 | 24.34 | 6.67 | 0.00 | 49.02 | 61.78 | 28.86 |

**Impact of Learning Method Diversity.** We explore the impact of learning-method diversity through two experiments. The first experiment (Table 4) uses Qwen-2.5-7B with five learning methods (four training-free base methods plus SFT) to examine whether training-based approaches complement training-free learning. The second experiment (Table 5) uses DeepSeek-V3.1 with eight

Table 5: Learning method analysis with 8 methods on 6 environments using DeepSeek-V3.1. Methods combine different signals, selection strategies , and components. Three upper bounds show performance under Best Selection, Pareto Selection, and all methods combined. Cost in USD per environment. Full environment names in Appendix A6.

| Method | Learning Subset | | | | | | Avg. | |
|---|---|---|---|---|---|---|---|---|
| | 1-ID | 19-AS | 21-WN | 24-MM | 26-TA | 33-AD | Acc | Cost |
| *IO* | | | | | | | | |
| DeepSeek-V3.1 | 28.58 | 32.18 | 20.00 | 0.00 | 62.73 | 66.45 | 34.99 | 0.60 |
| Claude-4-sonnet | 18.22 | 42.32 | 13.33 | 0.00 | 81.67 | 73.46 | 38.17 | - |
| *Best Selection* | | | | | | | | |
| Dynamics + PO | **42.92** | **34.99** | 10.00 | 0.00 | 62.73 | **81.64** | 38.71 | 0.67 |
| Instruction + PO | 28.58 | 32.29 | 20.00 | 0.00 | 52.27 | 74.96 | 34.68 | 0.68 |
| Dynamics + Agent | 37.20 | 31.92 | 20.00 | 0.00 | 81.81 | 73.68 | 40.77 | 0.49 |
| Instruction + Agent | 28.58 | 30.53 | 20.00 | 0.00 | 62.73 | 73.70 | 35.92 | 0.60 |
| *Pareto Selection* | | | | | | | | |
| Dynamics + PO | 32.50 | 32.97 | **30.00** | 0.00 | 70.45 | 64.50 | 38.40 | 0.69 |
| Instruction + PO | 38.06 | 32.42 | **30.00** | 0.00 | **88.52** | 68.94 | **42.99** | 0.76 |
| Dynamics + Agent | 30.48 | 28.27 | 20.00 | 0.00 | 62.73 | 81.11 | 37.10 | **0.43** |
| Instruction + Agent | 32.13 | 32.18 | 10.00 | 0.00 | 62.73 | 62.74 | 33.30 | 0.53 |
| UpperBound (best) | 42.92 | 34.99 | 20.00 | 0.00 | 81.81 | 81.64 | 43.56 | – |
| UpperBound (pareto) | 38.06 | 32.97 | 30.00 | 0.00 | 88.52 | 81.11 | 45.11 | – |
| UpperBound (all) | 42.92 | 34.99 | 30.00 | 0.00 | 88.52 | 81.64 | 46.34 | – |

methods to assess how expanding the learning-method space affects the upper bound. We approximate this upper bound as the maximum performance over all methods per environment (UpperBound (all)). UpperBound (best) and UpperBound (pareto) denote upper bounds restricted to subsets that use only Best Selection or Pareto Selection, respectively.

Results reveal strong environment-method interactions. As shown in Table 4 and Table 5, Environment 1-ID consistently favors Dynamics + Agent across both base models, while Environment 26-TA performs optimally with Instruction + Agent. Even SFT, which achieves the best average performance (25.09%) in Table 4, is outperformed by other methods on specific environments. More critically, learning methods can produce negative outcomes when mismatched: Dynamics + Agent under Pareto Selection actually drops below baseline on Environment 19-AS. This validates that inappropriate learning strategies may harm performance. The environments also show varying sensitivity to learning methods: Environment 24-MM remains challenging across all configurations (0%), while Environment 26-TA exhibits dramatic variation (52%-88% depending on method). These patterns demonstrate that optimal learning strategies are environment-specific rather than universal, and that heterogeneous environments provide diverse learning signal characteristics.

The learning upper bound consistently achieves performance gains, though with diminishing returns as the method space expands. With five methods on Qwen-2.5-7B, the upper bound is 3.77 points higher than the best single method. On DeepSeek-V3.1, using eight methods raises the upper bound by 3.35 points over the best single method (from 42.99% to 46.34%). Most of this improvement already appears with four methods and expanding to eight methods adds only 1.23 points. This suggests that while adding methods consistently improves the upper bound, the marginal gains decrease, indicating that having effective methods matters more than having many methods.

**Impact of Environment Diversity.** We scale the learning experiments to all 36 environments using Gemini-2.5-Flash with 4 learning methods under Best Selection. As shown in Appendix A9, the best single method (Dynamics + Prompt) achieves 42.40%, only 3.0 points above baseline (39.41%). This marginal gain is significantly smaller than the improvements observed on the 6-environment subset, where single methods achieved up to 7.2% gains. The diminishing returns suggest that as environment heterogeneity increases, no single learning strategy can effectively handle all cases.

Figure 3 (left) further illustrates this limitation. When environments are sorted by performance gain, single learning methods show consistently declining benefits: high gains on a few environments,

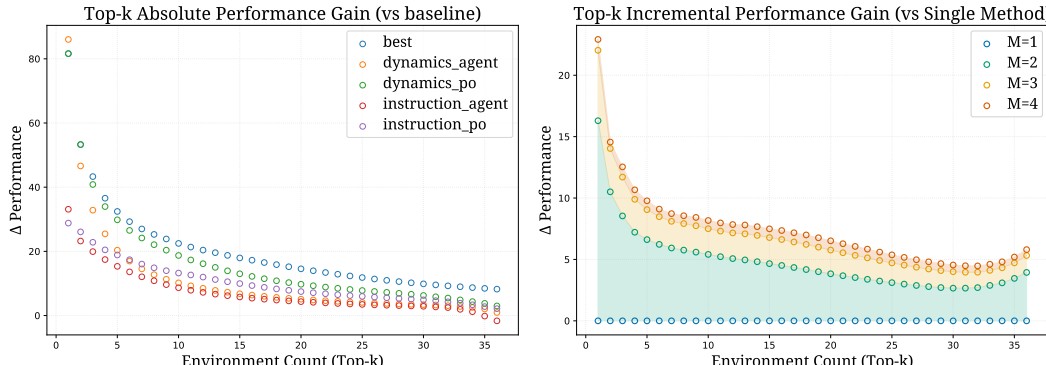

Figure 3: Impact of environment diversity on learning performance across 36 environments. Left: Performance gain of individual learning methods decreases as more environments are included (top-k by gain). Right: Expanding learning method space (M=1 to M=4) improves the upper bound with diminishing marginal returns. For each M, we evaluate all $\binom{4}{M}$ method subsets by taking the per-environment maximum over methods in each subset, averaging across environments, then averaging over subsets. This shows the performance upper bound achievable with M method choices.

but minimal or negative effects on many others. This finding confirms that different environments require different learning methods, and the mismatch penalty grows as environment diversity scales.

Despite these limitations, the upper bound (47.75%) maintains substantial potential for environment-adaptive learning. Compared to baseline, the upper bound achieves an 8.34% improvement (21% relative gain). More importantly, the 5.35% gap between upper bound and the best single method (42.40%) demonstrates significant room for adaptive selection. Figure 3 (right) shows that expanding from M=1 to M=4 methods consistently raises the upper bound, though with diminishing marginal gains, with the largest improvement coming from M=1 to M=2.

These findings point to an important direction for future work: automatic learning strategy design for heterogeneous environments. While our manually designed methods demonstrate the value of adaptive selection, the gap between single-method and upper-bound performance suggests substantial room for improvement. True adaptive learning requires systems that can automatically discover and compose environment-specific learning strategies, moving beyond hand-crafted approaches.

## 5.4 CASE STUDY

We present the environment code, observation code, reward computation, and agent instructions for the generated environment in Appendix B and learning case, prompts, agent structures, trajectories analysis in Appendix D.

## 6 CONCLUSION

We propose AUTOENV, an automated environment generation framework that alleviates environment scarcity for agent development by producing executable, validated environments at low cost. Using AUTOENV, we build AUTOENV-36, a benchmark of 36 heterogeneous environments with 358 validated levels, on which strong language models achieve only moderate but clearly stratified performance, showing that the benchmark is both challenging and diagnostic. We further formalize agent learning as a component-centric process over selection strategies, optimization schemes, and target components, which turns learning methods themselves into a search space. In this space we instantiate eight concrete learning methods and define a Learning Upper Bound that represents the best performance attainable when each environment can pick its own method. Experiments show that single fixed learning methods fail to scale across diverse environments, while environment-adaptive selection recovers a significant portion of the upper bound yet still leaves a noticeable gap. This gap highlights the need for richer learning method spaces and more powerful controllers that can design learning strategies automatically.

## ETHICS STATEMENT

We have read and will adhere to the ICLR Code of Ethics and the ICLR Code of Conduct. Our research introduces AutoEnv, a framework for LLM-powered environment generator. The methods used in our study are well-established for academic research. These environments do not contain any personally identifiable information (PII) or sensitive real-world data. Our work did not involve human subjects, crowd-sourcing, or the scraping of private data; therefore, Institutional Review Board (IRB) approval was not required.

We acknowledge that research on autonomous agents carries potential dual-use risks. To mitigate these, our experiments are intentionally confined to benign, closed-world tasks such as online shopping and household activities within simulated settings. We followed good scholarly practice by reporting our methods and results transparently and citing prior work accurately. The authors declare no competing interests or external sponsorships that could have influenced the outcomes of this research.

## REPRODUCIBILITY STATEMENT

We are committed to ensuring the reproducibility of our research. All essential details for reproducing our results are provided within this paper. The AUTOENV framework architecture, including the three-layer abstraction (BaseEnv, ObsEnv, SkinEnv) and the domain-specific language (DSL) specifications, are detailed in Section 3 and Appendix B. The complete AUTOENV-36 dataset statistics, environment features, and validation procedures are described in Section 3.3 and Appendix C. Our experimental setup, including the specific language models used (GPT-4o-mini, GPT-5, O3, Claude-4-Sonnet, Kimi-K2, DeepSeek-V3.1, Gemini-2.5-Flash), learning method configurations, temperature settings, and evaluation protocols, is described in Section 5.1. The learning strategy formalization with all component types (selection strategies, optimization signals, target components) and their implementations are provided in Section 4 and Appendix E. To facilitate full replication of our environment generation pipeline and learning adaptation experiments, we will release our complete codebase, generated environments, and evaluation scripts as supplementary material.

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

## A  ENVIRONMENT ABSTRACTION

**Environment File**

```python
class BaseEnv(ABC):
    """Defines the true state, transition, and reward."""
    def __init__(self, env_id: int):
        self.env_id = env_id # env_id means the id of this class env.
        self._t = 0
        self._history: List = [] # past state
        self._state = None # current state
        self.configs = None
        # Optional: store latest action side-effect/result for UI/agent feedback
        self._last_action_result: Any = None
        self._dsl_config()

    @abstractmethod
    def _dsl_config(self):
        """
        Load DSL configuration from YAML file.
        Expected path: worlds/{env_id}/config.yaml
        """
        pass

    @abstractmethod
    def reset(self, mode: str = "load",
        world_id: Optional[str] = None,
        seed: Optional[int] = None):
        """
        Reset environment by either loading an existing world or generating a new one.

        Args:
            mode: "load" to load from file, "generate" to generate a new world
            world_id: Used only in "load" mode. Load the world with this id.
            seed: Used only in "generate" mode. Generate a new world with this seed.

        Behavior:
            - If mode == "load": Load world state from file using world_id.
            - If mode == "generate": Generate new world using seed, then load it.
        """
        pass

    @abstractmethod
    def _load_world(self, world_id: str) -> Dict[str, Any]:
        """
        Load world state from file.

        Args:
            world_id: Identifier of the world file to load

        Returns:
            Complete world state dictionary
        """
        pass

    @abstractmethod
    def _generate_world(self, seed: Optional[int] = None) -> str:
        """
        Generate complete world using generator pipeline and save to file.

        Args:
            seed: Random seed for reproducible generation

        Returns:
            world_id: Identifier of the generated world file
        """
        pass

    @abstractmethod
    def transition(self, action: Dict[str, Any]) -> Dict[str, Any]:
        """
        State transition function.
        Input an action dict with two key:
        - action: str, the name of action
        - params: dict, the parameters of action
        And then apply the transition to self.state
        """
```

```python
            pass

        @abstractmethod
        def reward(self, action: Dict[str, Any]) -> Tuple[float, List[str], Dict[str, Any]]:
            """
            Reward Function.
            It define agent how to get a reward.
            The state can be obtained from self.state,
            and past state can be gained from self.history.
            """
            pass

    class ObsEnv(BaseEnv):
        """Adds observation interface: output semantic observation from true state."""

        def __init__(self, env_id, obs_policy: ObservationPolicy):
            super().__init__(env_id)
            self.obs_policy = obs_policy

        @abstractmethod
        def observe_semantic(self) -> Dict[str, Any]:
            """
            Semantic-level observation.
            The observation policy refer to the observation state,
            such as full, partial, radius.
            And this function is used to transfer state to semantic obs.
            """
            pass

    class SkinEnv(ObsEnv):
        """Adds rendering interface: semantic observation -> final input (X)."""

        @abstractmethod
        def render_skin(self, omega: Dict[str, Any]) -> Any:
            """Render the final input from semantic observation."""
            pass

        def done(self) -> bool:
            # Default: only step count; override/add conditions if needed
            return self._t >= self.configs["termination"]["max_steps"]

        def step(self, action: Dict[str, Any]):
            """
            Basic step logic for an environment
            You can modify it in anywhere you want.
            """
            # Reset last action result; transition can set it
            self._last_action_result = None
            s_next = self.transition(action)
            reward, events, rinfo = self.reward(action)
            self._t += 1
            raw_obs = self.observe_semantic()
            agent_obs = self.render_skin(raw_obs)
            if_done = self.done(s_next)
            info = {
                "raw_obs": raw_obs,
                "skinned": agent_obs,
                "events": events,
                "reward_info": rinfo,
                "last_action_result": self._last_action_result,
            }
            return s_next, reward, if_done, info
```

---

**Algorithm 1** AUTOENV: Automated Environment Generation

---

**Require:** Environment theme $\theta$, language models $LM_{exec}$, $LM_{reflect}$
**Ensure:** Validated environment $\mathcal{E} = (\mathcal{S}, \mathcal{A}, T, R, \Omega, \tau)$ with up to 15 validated levels
1: $D \leftarrow$ DESIGNAUTHORING$(\theta, LM_{exec})$      $\triangleright$ Generate structured environment design
2: $config \leftarrow$ DSLSYNTHESIS$(D, LM_{exec})$      $\triangleright$ Convert design to DSL YAML
3: VALIDATEDSL$(config)$      $\triangleright$ Check schema and interface alignment
4:
5: $(files, generator, validator) \leftarrow$ CODESYNTHESIS$(config, LM_{exec})$    $\triangleright$ Generate BaseEnv/ObsEnv/SkinEnv, level generator, and validator
6:             $\triangleright$ **Self-repair on code (up to 40 steps)**
7: **for** $t = 1$ to 40 **do**
8:   **if** BASICCODETEST$(files)$ **then**      $\triangleright$ Import module, reset and step env, call generator/validator
9:    **break**
10:   **else**
11:    $files \leftarrow$ SELFREPAIR$(files, LM_{reflect})$     $\triangleright$ Edit code based on error messages
12:   **end if**
13: **end for**
14: **if** $\neg$BASICCODETEST$(files)$ **then**
15:   **reject** environment
16: **end if**
17:        $\triangleright$ **Execution: runtime stability test with a small ReAct agent**
18: **if** $\neg$EXECUTIONTEST$(files)$ **then**
19:   **reject** environment
20: **end if**
21:       $\triangleright$ **Level Generation: generate levels and compute upper bounds**
22: $levels \leftarrow \emptyset$, $bounds \leftarrow \emptyset$
23: **for** $i = 1$ to 15 **do**
24:   $level_i \leftarrow$ GENERATELEVEL$(generator, config)$
25:   **if** VALIDATELEVEL$(level_i, validator)$ **then**
26:    $b_i \leftarrow$ COMPUTEUPPERBOUND$(level_i, validator)$    $\triangleright$ max_reward-style upper bound
27:    $levels \leftarrow levels \cup \{level_i\}$,   $bounds \leftarrow bounds \cup \{b_i\}$
28:   **end if**
29: **end for**
30: **if** $|levels| = 0$ **then**
31:   **reject** environment
32: **end if**
33:        $\triangleright$ **Reliability: differential model testing on rewards**
34: **if** $\neg$CONSISTENCYCHECK$(levels, \text{GPT-4o-mini}, \text{DeepSeek-V3.1})$ **then**
35:   **reject** environment
36: **end if**
37:
38: SAVECONFIGURATION$(levels, bounds, config)$
39: $\mathcal{E} \leftarrow$ PACKAGEENVIRONMENT$(levels, bounds, files)$
40: **return** $\mathcal{E}$

---

Table A6: Environment features, only main feature are shown in this table.

| EnvName | Observation | Reward | Semantic |
|---|---|---|---|
| 1_InterDimension | Partially Observable | Accum | Aligned |
| 2_BioLumine | Partially Observable | Accum | Aligned |
| 3_BackwardTimes | Partially Observable | Binary | Aligned |
| 4_MagneticField | Fully Observable | Accum | Aligned |
| 5_TerraForming | Fully Observable | Accum | Aligned |
| 6_EntropyReversal | Fully Observable | Accum | Aligned |
| 7_QuantumMaze | Partially Observable | Binary | Aligned |
| 8_MolecularTaste | Partially Observable | Binary | Aligned |
| 9_CollectiveConsciousness | Fully Observable | Accum | Aligned |
| 10_WeatherControl | Partially Observable | Accum | Aligned |
| 11_UndergroundCity | Fully Observable | Accum | Aligned |
| 12_SentientArchitecture | Fully Observable | Accum | Aligned |
| 13_OpticalAnalysis | Fully Observable | Binary | Aligned |
| 14_FieldDetection | Partially Observable | Binary | Aligned |
| 15_SystemEngineering | Fully Observable | Binary | Aligned |
| 16_GearOptimization | Fully Observable | Binary | Aligned |
| 17_LabExperimentation | Fully Observable | Accum | Inverse |
| 18_LifeSimulation | Partially Observable | Accum | Aligned |
| 19_AgriculturalSimulation | Partially Observable | Accum | Inverse |
| 20_GridNavigation | Partially Observable | Binary | Aligned |
| 21_WorldNavigation | Partially Observable | Binary | Inverse |
| 22_ColumnStrategy | Fully Observable | Binary | Aligned |
| 23_DangerDetection | Partially Observable | Binary | Inverse |
| 24_MemoryMatching | Partially Observable | Binary | Aligned |
| 25_PatternMatching | Partially Observable | Accum | Inverse |
| 26_TileArrangement | Fully Observable | Accum | Aligned |
| 27_TerrainNavigation | Partially Observable | Binary | Aligned |
| 28_IcyNavigation | Partially Observable | Binary | Inverse |
| 29_LogisticsPuzzle | Fully Observable | Binary | Inverse |
| 30_ObjectManipulation | Partially Observable | Accum | Aligned |
| 31_PatternCompletion | Partially Observable | Accum | Aligned |
| 32_DreamSequence | Partially Observable | Binary | Aligned |
| 33_AlienDecision | Partially Observable | Accum | Inverse |
| 34_ShadowPuppet | Fully Observable | Binary | Aligned |
| 35_BattlefieldTactics | Partially Observable | Accum | Aligned |
| 36_WarehousePuzzle | Fully Observable | Binary | Aligned |

Table A7: Capability coverage of AUTOENV-36. Columns: Nav = Navigation / Spatial; POM = Partial Observability & Memory; Inv. = Counterintuitive / Inverted; Ctrl/Res = Control / Resource / Multi-Objective; Patt = Pattern / Symbolic; Plan = Planning / Long-horizon; Multi = Multi-agent / Adversarial.

| EnvName | Nav | POM | Inv. | Ctrl/Res | Patt | Plan | Multi |
|---|---|---|---|---|---|---|---|
| 1_InterDimension | | ✓ | | ✓ | ✓ | ✓ | |
| 2_BioLumine | | ✓ | | | ✓ | ✓ | |
| 3_BackwardTimes | | ✓ | | | ✓ | ✓ | |
| 4_MagneticField | | | | | ✓ | ✓ | |
| 5_TerraForming | | | | ✓ | | ✓ | |
| 6_EntropyReversal | | | | ✓ | | ✓ | |
| 7_QuantumMaze | ✓ | ✓ | | | | ✓ | |
| 8_MolecularTaste | ✓ | ✓ | | | ✓ | ✓ | |
| 9_CollectiveConsciousness | | | | ✓ | | ✓ | |
| 10_WeatherControl | | ✓ | ✓ | ✓ | | ✓ | |
| 11_UndergroundCity | ✓ | | ✓ | ✓ | | ✓ | |
| 12_SentientArchitecture | | | | ✓ | | ✓ | ✓ |
| 13_OpticalAnalysis | | | | | ✓ | ✓ | |
| 14_FieldDetection | ✓ | ✓ | | | ✓ | ✓ | |
| 15_SystemEngineering | | | | ✓ | ✓ | ✓ | |
| 16_GearOptimization | | | | | ✓ | ✓ | |
| 17_LabExperimentation | | ✓ | ✓ | ✓ | ✓ | ✓ | |
| 18_LifeSimulation | ✓ | ✓ | | ✓ | | ✓ | |
| 19_AgriculturalSimulation | ✓ | ✓ | ✓ | ✓ | | ✓ | |
| 20_GridNavigation | ✓ | ✓ | | | | ✓ | |
| 21_WorldNavigation | ✓ | ✓ | ✓ | | | ✓ | |
| 22_ColumnStrategy | | | | | ✓ | ✓ | ✓ |
| 23_DangerDetection | ✓ | ✓ | ✓ | | | ✓ | |
| 24_ChaosSlidePuzzle | | | | | ✓ | ✓ | |
| 25_MemoryPairMatching | | ✓ | | | ✓ | ✓ | |
| 26_MismatchedMemoryGame | | ✓ | ✓ | | ✓ | ✓ | |
| 27_TerrainNavigation | ✓ | ✓ | | | | ✓ | |
| 28_IcyNavigation | ✓ | ✓ | ✓ | | | ✓ | |
| 29_LogisticsPuzzle | ✓ | | ✓ | | ✓ | ✓ | |
| 30_ObjectManipulation | ✓ | ✓ | | ✓ | | ✓ | |
| 31_PatternCompletion | | | | | ✓ | ✓ | |
| 32_DreamSequence | ✓ | ✓ | ✓ | | | ✓ | |
| 33_AlienDecision | | ✓ | ✓ | ✓ | | ✓ | |
| 34_ShadowPuppet | ✓ | | | | ✓ | ✓ | |
| 35_BattlefieldTactics | ✓ | ✓ | | ✓ | | ✓ | ✓ |
| 36_WarehousePuzzle | ✓ | | | | ✓ | ✓ | |

# B ENVIRONMENT GENERATION DETAILS

## B.1 ENVIRONMENT GENERATION ALGORITHM

## B.2 ENVIRONMENT FEATURES

## B.3 ENVIRONMENT FILELIST

**Environment File**

```
Environment/
  |-- action_space.txt
  |-- agent_instruction.txt
  |-- config.yaml
  |-- env_desc.txt
  |-- env_generate.py
  |-- env_implement.txt
  |-- env_main.py
  |-- env_main_use.py
  |-- env_obs.py
  |-- env_validator.py
  |-- level_max_rewards.json
  |-- levels/
  |   |-- level_01.yaml
  |   |-- level_02.yaml
  |   |-- ...
  \-- val_levels/
      |-- level_11.yaml
      |-- level_12.yaml
      |-- ...
```

## B.4 ENVIRONMENT DSL

**22_ColumnStrategy Config**

```yaml
meta:
  id: "tower_stack_connect_four"
  name: "Tower-Stack Connect-Four"
  description: "Strategic Connect-Four game where agent competes against heuristic
  opponent on 6x7 grid"

state_template:
  globals:
    max_steps: 40
    board_height: 6
    board_width: 7
  agent:
    player_id: 1
    wins: 0
  opponent:
    player_id: 2
    last_move: -1
    policy: "heuristic_depth1"
  board:
    grid: []
    filled_columns: []
  game:
    current_player: 1
    winner: 0
    game_over: false
    moves_made: 0

observation:
  policy: "full_board"
  params: {}
  expose:
    - board.grid
    - opponent.last_move
    - globals.max_steps
    - game.moves_made
```

```
         - t

     reward:
       events:
         - trigger: "game_won"
           value_key: "win_rewards"
         - trigger: "game_lost"
           value_key: "loss_rewards"
         - trigger: "game_timeout"
           value_key: "timeout_rewards"
       win_rewards:
         agent_victory: 1.0
       loss_rewards:
         opponent_victory: 0.0
       timeout_rewards:
         no_winner: 0.0

     transition:
       actions:
         - name: "drop_disk"
           params: [column]

     skin:
       type: "text"
       template: |
         Step {t}/{max_steps} | Moves: {moves_made}
         Last opponent move: Column {opponent_last_move}

         Board (1=You, 2=Opponent, 0=Empty):
         {board_display}

         Available actions: drop_disk(column) where column in [0,1,2,3,4,5,6]
         Game status: {game_status}

     termination:
       max_steps: 40
       conditions:
         - "game.game_over == true"
         - "game.winner != 0"

     generator:
       mode: "procedural"
       output_format: "yaml"
       pipeline:
         - name: "init_from_template"
           desc: "Initialize world with empty 6x7 Connect-Four board"
           args: {}
         - name: "setup_empty_board"
           desc: "Create 6x7 grid filled with zeros, initialize column tracking"
           args:
             height: 6
             width: 7
         - name: "initialize_game_state"
           desc: "Set agent as first player, reset counters and flags"
           args:
             starting_player: 1
         - name: "setup_opponent_heuristic"
           desc: "Configure opponent AI with depth-1 heuristic policy"
           args:
             policy_type: "win_block_random"
             depth: 1

       randomization:
         seed_based: true
         parameters:
           opponent_randomness: [0.0, 0.1]

     world_loading:
       directory: "worlds/{env_id}/"
       format: "yaml"
       validation_schema: "state_template"
       naming_convention: "{world_id}.yaml"

     misc:
       logging: true
       store_rollouts: true
       debug_mode: false
```

## B.5 Environment Code

**22_ColumnStrategy Main**

```python
class ConnectFourOpponent:
    """
    Opponent policy:
    1) Try winning immediately
    2) Otherwise block agent's winning move
    3) Otherwise choose random valid column
    """

    @staticmethod
    def get_move(board_grid):
        # Try winning
        col = ConnectFourOpponent.check_winning_move(board_grid, player=2)
        if col != -1: return col

        # Try blocking
        col = ConnectFourOpponent.check_winning_move(board_grid, player=1)
        if col != -1: return col

        # Random fallback
        return ConnectFourOpponent.get_random_move(board_grid)

    @staticmethod
    def check_winning_move(board_grid, player): ...
    @staticmethod
    def get_random_move(board_grid): ...
    @staticmethod
    def check_win(board_grid, row, col, player): ...

class ConnectFourEnv(SkinEnv):
    """
    Connect-Four environment definition:
    - Supports world loading & generation
    - Agent vs Opponent dynamics
    - Reward calculation and rendering
    """

    def __init__(self, env_id):
        self.obs_policy = ConnectFourObservation()
        super().__init__(env_id, self.obs_policy)
        self.generator = ConnectFourGenerator(env_id, self.configs)

    def reset(self, mode="generate", world_id=None, seed=None):
        if mode == "load":
            self._state = self._load_world(world_id)
        elif mode == "generate":
            world_id = self._generate_world(seed)
            self._state = self._load_world(world_id)
        self._t, self._history = 0, []
        return self._state

    def transition(self, action):
        """
        Agent move:
            - Drop disk into column
            - Check win → mark game_over
        Opponent move (if game not over):
            - Policy selects column
            - Drop disk
            - Check win → mark game_over
        """
        ...

    def reward(self, action):
        """
        Reward rules:
        - Agent win → 1.0 + "game_won"
        - Opponent win → 0.0 + "game_lost"
        - Timeout → 0.0 + "game_timeout"
        """
        ...

    def observe_semantic(self):
```

```
        """Return structured state for agent (board, moves, etc.)."""
        ...

    def render_skin(self, omega):
        """
        Render board to text:
        - Show step count, last move
        - Board as grid (1=Agent, 2=Opponent, 0=Empty)
        - Game status (In progress / Win / Loss)
        """
        ...

    def done(self, state=None):
        """Episode ends on win/loss/timeout."""
        return (self._state["game"]["game_over"]
                or self._t >= self.configs["termination"]["max_steps"])
```

### 22_ColumnStrategy Observation

```python
class ConnectFourObservation(ObservationPolicy):
    def __call__(self, env_state: Dict[str, Any], t: int) -> Dict[str, Any]:
        return {
            'board_grid': env_state['board']['grid'],
            'opponent_last_move': env_state['opponent']['last_move'],
            'max_steps': env_state['globals']['max_steps'],
            'moves_made': env_state['game']['moves_made'],
            't': t + 1
        }
```

## B.6 ENVIRONMENT INSTRUCTION

### 22_ColumnStrategy Agent Instruction

```
You are navigating a deceptive 10x10 grid world where appearances may not match
underlying behavior. Your goal is to locate and step on the single true goal tile within
30 steps.

Observations and movement:
- You start on a random floor tile and can see a 5x5 area around yourself.
- Use movement actions (MoveNorth, MoveSouth, MoveEast, MoveWest) or Wait to pause.

Guidance:
- Symbols and visuals may not reliably indicate how tiles behave.
- Use cautious exploration and simple empirical tests to verify which tiles are safe,
blocked, or special.
- Systematically probe candidate goal tiles to identify the true objective while managing
the step limit.
```

### 22_ColumnStrategy Action Space

```json
[
  {
    "name": "MoveNorth",
    "description": "Move the agent one step north (up) on the grid.",
    "parameters": {}
  },
  {
    "name": "MoveSouth",
    "description": "Move the agent one step south (down) on the grid.",
    "parameters": {}
  },
  {
    "name": "MoveEast",
    "description": "Move the agent one step east (right) on the grid.",
    "parameters": {}
  },
  {
```

```
    "name": "MoveWest",
    "description": "Move the agent one step west (left) on the grid.",
    "parameters": {}
  },
  {
    "name": "Wait",
    "description": "Stay in the current position without moving.",
    "parameters": {}
  }
]

Actions should be formatted as dictionaries with an 'action' key specifying the action
name and a 'params' key containing the required parameters as a dictionary.
For example: "action": "ACTION_NAME", "params": "param1": value1, "param2": value2}}
```

## B.7 HUMAN REVIEW ON ENVIRONMENT THEME

**Initial Theme**

```
Dream Sequence Navigation
The agent is trapped in a surreal dream, moving between strange rooms connected by
↪   colored doors. Some rooms have weird physics like anti gravity or time distortion.
↪   The agent needs to find a  wake up portal by following the right sequence of rooms
↪   under a step limit. There might be a special key in one of the rooms that is
↪   required to activate the portal. The rules should be consistent so that an agent
↪   can learn them over time.
```

**Theme after human review**

```
Dream Sequence Navigation Environment

## Core Concept
The agent explores a simple dreamscape where dream logic replaces normal physics. Each
↪   room has unique properties, but the rules are consistent and learnable across
↪   episodes. The agent must find the correct sequence of rooms to reach the Awakening
↪   Portal.

## Objective
Find and reach the "Awakening Portal" within 40 steps by navigating through the correct
↪   sequence of dream rooms.

## State (Agent Observation)
1. Current room ID (0-5 for Easy, 0-7 for Medium, 0-9 for Hard).
2. Available exits: up to 3 colored doors (Red, Blue, Green).
3. Room special property: Normal, Anti-Gravity, or Time-Slow.
4. Dream key collected: Yes/No (only one key exists).
5. Steps remaining (0-40).

## Actions
0. EnterRedDoor
1. EnterBlueDoor
2. EnterGreenDoor
3. PickUpKey (if key is present in current room)
4. Wait

Invalid actions (entering non-existent doors) waste the step but keep the agent in the
↪   same room.

## Key Mechanisms

Room Navigation
- Each room has 1-3 colored doors leading to other rooms.
- Door connections are fixed but not intuitive (Red door might lead backwards, Blue
↪   door might skip rooms).
- Room layout is the same across all episodes.
- Only the starting room varies.

Dream Logic Rules
- Anti-Gravity rooms: Red and Blue doors swap destinations.
```

```
- Time-Slow rooms: The agent must Wait one turn before any door becomes usable.
- Normal rooms: Doors work as expected.

Portal Mechanics
- The Awakening Portal appears only in the final room (room ID varies by difficulty).
- The portal only becomes active if the agent has collected the dream key.
- The key appears in exactly one random room per episode.

Win Condition Discovery
The agent must learn:
1) Which room contains the key.
2) Which room contains the portal.
3) The correct path between them.

Path length:
- 4-6 rooms for Easy.
- 6-8 rooms for Medium.
- 8-10 rooms for Hard.

## Reward Structure
Reward type: Binary (0/1).
- +1: The agent reaches the activated Awakening Portal with the key before step 40.
- 0: All other outcomes.

## Termination Conditions
1. Success: Portal reached with key.
2. Step limit: 40 steps elapsed.
3. Dead end: The agent enters a room with no exits (immediate failure).

## Special Requirements
1. Consistent rules: Door color mappings and room effects never change between
↪ episodes.
2. Key location: Randomized each episode but always appears in exactly one room.
3. Portal location: Fixed final room per difficulty level, but the path to reach it
↪ varies.
4. Difficulty scaling: Only the number of rooms increases (6/8/10); all other mechanics
↪ are identic
```

Before code generation, we perform a human review step that turns high level, often vague themes into precise environment specifications. In both the automated and supervised settings, the initial themes are written by a language model. For the 25 supervised themes, a human reviewer then inspects the LLM-generated theme, identifies ambiguities or missing details, and writes concrete revision instructions, for example asking to make state, action, reward, and difficulty rules explicit or to resolve redundant or conflicting descriptions. These instructions are passed back to the language model, which rewrites the theme into a more precise, code ready specification that still preserves the original idea. For instance, in the *Dream Sequence Navigation* environment, the original theme only mentioned a dream world with strange rooms and a wake up portal, while the reviewed version fixes the observation fields, action space, dream logic rules, key and portal mechanics, and reward and termination conditions. This review process reduces rule inconsistencies caused by overly abstract themes and directly leads to the higher overall success rate reported for supervised themes.

## C ENVIRONMENT EVELUATION DETAILS

### C.1 METRICS

**22_ColumnStrategy level_max_rewards**

```
{
  "environment_id": "string",
  "calculation_timestamp": "YYYY-MM-DD HH:MM:SS",
  "levels": {
    "level_x.yaml": {
      "max_reward": "<float>",
      "calculation_method": "string",
      "notes": "string",
      "board_empty": "<bool>",
      "initial_moves": "<int>",
      "game_over": "<bool>"
```

```
1350        },
1351        "...": { "...": "..." }
1352      },
          "summary": {
1353        "total_levels": "<int>",
1354        "average_max_reward": "<float>",
1355        "min_max_reward": "<float>",
            "max_max_reward": "<float>",
1356        "methodology": "string"
1357      }
          }
1358
```

1359

## 22 ColumnStrategy Validator

1360
1361
```python
1362    class ConnectFourValidator:
1363        """
           Validator for Connect-Four environment levels.
1364        Ensures solvability, proper rewards, and valid board state.
           """
1365
1366        def __init__(self):
1367            self.board_height = 6
               self.board_width = 7
1368            self.max_steps = 40
1369
           def validate_level(self, level_state: Dict[str, Any]) -> Tuple[bool, List[str]]:
1370            issues = []
1371            issues.extend(self._check_level_solvability(level_state))
               issues.extend(self._validate_reward_structure(level_state))
1372            issues.extend(self._validate_basic_state(level_state))
1373            return len(issues) == 0, issues
1374
           # === Solvability Checks ===
1375
           def _check_level_solvability(self, level_state: Dict[str, Any]) -> List[str]:
1376            issues = []
1377            issues.extend(self._analyze_action_constraints(level_state))
               issues.extend(self._check_target_reachability(level_state))
1378            issues.extend(self._check_impossible_patterns(level_state))
1379            return issues
1380
           def _analyze_action_constraints(self, level_state: Dict[str, Any]) -> List[str]:
1381            """Check board exists, dimensions correct, and moves are possible."""
               issues = []
1382            grid = level_state.get("board", {}).get("grid")
1383            if grid is None:
                   issues.append("Missing board grid")
1384                return issues
1385            if len(grid) != self.board_height or len(grid[0]) != self.board_width:
                   issues.append("Invalid board dimensions")
1386            if all(col[0] != 0 for col in zip(*grid)):
                   issues.append("No available columns to play")
1387            return issues
1388
           def _check_target_reachability(self, level_state: Dict[str, Any]) -> List[str]:
1389            """Verify game not already lost and steps are sufficient for a win."""
               issues = []
1390            game = level_state.get("game", {})
1391            if game.get("game_over", False):
                   issues.append("Game already finished")
1392            remaining_steps = level_state.get("globals", {}).
1393            get("max_steps", self.max_steps) \
1394                            - game.get("moves_made", 0)
               if remaining_steps <= 0:
1395                issues.append("No remaining steps to play")
1396            return issues
1397
           def _check_impossible_patterns(self, level_state: Dict[str, Any]) -> List[str]:
1398            """Detect floating pieces, unbalanced counts, or invalid disk numbers."""
               issues = []
1399            # Example: disk count difference should never exceed 1
1400            grid = level_state.get("board", {}).get("grid", [])
1401            agent_count = sum(row.count(1) for row in grid)
               opp_count = sum(row.count(2) for row in grid)
1402            if abs(agent_count - opp_count) > 1:
1403                issues.append("Unrealistic disk count")
```

```python
        return issues

    # === Reward & State Checks ===

    def _validate_reward_structure(self, level_state: Dict[str, Any]) -> List[str]:
        """Check binary reward and termination condition."""
        issues = []
        game = level_state.get("game", {})
        if "winner" not in game or "game_over" not in game:
            issues.append("Missing game outcome fields")
        if "max_steps" not in level_state.get("globals", {}):
            issues.append("Missing max_steps termination")
        return issues

    def _validate_basic_state(self, level_state: Dict[str, Any]) -> List[str]:
        """Check required keys and valid cell values."""
        issues = []
        for key in ["globals", "agent", "opponent", "board", "game"]:
            if key not in level_state:
                issues.append(f"Missing key: {key}")
        grid = level_state.get("board", {}).get("grid", [])
        for r, row in enumerate(grid):
            for c, cell in enumerate(row):
                if cell not in [0, 1, 2]:
                    issues.append(f"Invalid cell value {cell} at ({r}, {c})")
        return issues

def validate_connect_four_level(level_state: Dict[str, Any]) -> Tuple[bool, List[str]]:
    """Convenience wrapper."""
    return ConnectFourValidator().validate_level(level_state)
```

## C.2 DETAILED PERFORMANCE

Table A8: Detailed Experiments

| EnvName | Claude-4-Sonnet | DeepSeek-V3.1 | Gemini-2.5-Flash | GPT-4o-mini | GPT-5 | Kimi-k2 | O3 |
|---|---|---|---|---|---|---|---|
| 1_InterDimension | 18.22% | 22.14% | 28.11% | 13.34% | 138.84% | 6.53% | 53.84% |
| 2_BioLumine | 16.67% | 20.00% | 26.67% | 3.33% | 20.00% | 16.67% | 31.11% |
| 3_BackwardTimes | 26.67% | 26.67% | 0.00 | 0.00 | 0.00 | 3.33% | 30.00% |
| 4_MagneticField | 50.56% | 18.33% | 50.00% | 6.67% | 50.00% | 37.22% | 53.33% |
| 5_TerraForming | 44.00% | 46.28% | 43.07% | 28.71% | 30.33% | 40.00% | 30.78% |
| 6_EntropyReversal | 7.96% | 2.65% | 1.66% | 3.35% | 4.00% | 5.44% | 6.90% |
| 7_QuantumMaze | 3.33% | 3.33% | 6.67% | 0.00 | 10.00% | 3.33% | 6.67% |
| 8_MolecularTaste | 58.33% | 50.00% | 50.00% | 0.00 | 62.50% | 29.17% | 87.50% |
| 9_CollectiveConsciousness | 13.37% | 54.12% | 13.96% | 8.97% | 20.62% | 27.13% | 12.80% |
| 10_WeatherControl | 38.42% | 34.35% | 16.64% | 22.24% | 39.53% | 36.73% | 41.59% |
| 11_UndergroundCity | 46.72% | 42.07% | 41.27% | 10.49% | 44.21% | 40.37% | 36.56% |
| 12_SentientArchitecture | 36.70% | 35.06% | 33.21% | 18.18% | 32.03% | 32.85% | 35.73% |
| 13_OpticalAnalysis | 40.00% | 36.67% | 16.67% | 26.67% | 20.00% | 33.33% | 33.33% |
| 14_FieldDetection | 40.00% | 36.67% | 40.00% | 3.33% | 60.00% | 33.33% | 53.33% |
| 15_SystemEngineering | 66.67% | 43.33% | 86.67% | 6.67% | 100.00% | 36.67% | 100.00% |
| 16_GearOptimization | 96.67% | 93.33% | 100.00% | 13.33% | 100.00% | 93.33% | 100.00% |
| 17_LabExperimentation | 27.72% | 27.31% | 29.59% | 9.76% | 15.12% | 21.27% | 20.20% |
| 18_LifeSimulation | 11.89% | 11.04% | 8.43% | 3.15% | 14.54% | 11.07% | 14.47% |
| 19_AgriculturalSimulation | 42.32% | 33.02% | 33.51% | 20.92% | 32.04% | 29.11% | 34.31% |
| 20_GridNavigation | 40.00% | 30.00% | 53.33% | 10.00% | 60.00% | 26.67% | 73.33% |
| 21_WorldNavigation | 13.33% | 16.67% | 6.67% | 0.00 | 20.00% | 16.67% | 6.67% |
| 22_ColumnStrategy | 46.67% | 16.67% | 63.33% | 10.00% | 90.00% | 26.67% | 83.33% |
| 23_DangerDetection | 6.67% | 10.00% | 10.00% | 10.00% | 10.00% | 10.00% | 6.67% |
| 24_MemoryMatching | 0.00 | 0.00 | 0.00 | 0.00 | 0.00 | 0.00 | 10.00% |
| 25_PatternMatching | 89.39% | 89.77% | 90.91% | 13.64% | 90.91% | 87.88% | 90.91% |
| 26_TileArrangement | 81.67% | 65.80% | 83.18% | 46.89% | 40.80% | 61.25% | 68.33% |
| 27_TerrainNavigation | 90.00% | 73.33% | 93.33% | 20.00% | 100.00% | 66.67% | 100.00% |
| 28_IcyNavigation | 60.00% | 46.67% | 33.33% | 3.33% | 80.00% | 33.33% | 96.67% |
| 29_LogisticsPuzzle | 83.33% | 20.00% | 56.67% | 0.00 | 100.00% | 46.67% | 100.00% |
| 30_ObjectManipulation | 11.35% | 9.46% | 11.98% | 9.46% | 13.24% | 11.98% | 13.87% |
| 31_PatternCompletion | 42.09% | 23.84% | 32.40% | 41.34% | 21.79% | 20.11% | 34.82% |
| 32_DreamSequence | 96.67% | 86.67% | 90.00% | 40.00% | 100.00% | 90.00% | 93.33% |
| 33_AlienDecision | 73.46% | 65.61% | 67.52% | 26.96% | 64.33% | 80.68% | 67.30% |
| 34_ShadowPuppet | 23.33% | 13.33% | 50.00% | 0.00 | 10.00% | 10.00% | 30.00% |
| 35_BattlefieldTactics | 0.00 | 0.00 | 16.67% | 0.00 | 0.00 | 1.67% | 6.67% |
| 36_WarehousePuzzle | 26.67% | 20.00% | 33.33% | 0.00 | 90.00% | 6.67% | 90.00% |
| **Average** | 40.86% | 34.01% | 39.41% | 11.96% | 46.80% | 31.49% | 48.73% |

**Case study: normalized accuracy above 100% in INTERDIMENSION.** 1-INTERDIMENSION is a multi-dimensional currency arbitrage and risk-management environment where an agent operates over a fixed horizon (typically 40 steps) on three ledgers (Mass, Entropy, Historical). The agent starts with an inventory of items (e.g., artifacts, dark_matter, neural_matrices) and balances in the three dimension-specific currencies, and observes a noisy exchange-rate matrix. Available actions include PROPOSE_TRADE (exchanging items for currency across dimensions), CONVERT_CREDITS (converting currencies between dimensions), HEDGE (temporarily freezing volatility in a dimension), RESEARCH (improving the accuracy of some exchange-rate observations), and DONATE (reducing embargo risk by donating items). The core objective is to maximize total profit while avoiding debt or dimensional embargo, and to keep the ledgers stable and risk under control.

For benchmarking, each level is assigned a validator-estimated "maximum reward" produced by an environment analysis script that performs a heuristic arbitrage search based on the reward configuration. This estimator first computes a conservative upper bound on achievable net profit and then, under several behavioral assumptions (e.g., only a fraction of steps maintain low embargo risk or non-negative ledgers, and only a limited number of successful RESEARCH discoveries), adds stability, fairness, research, and goal bonuses. It is therefore designed as a convenient heuristic upper bound rather than a strict mathematical maximum under the actual reward function.

In contrast, the runtime reward is considerably more permissive: profit rewards accumulate every positive profit increment without subtracting subsequent losses, so sequences of alternating gains and losses can yield cumulative profit much higher than the final net wealth; as long as the agent avoids debt and keeps embargo risk below the thresholds, it can receive stability and fairness rewards on nearly every step; and frequent successful RESEARCH can trigger additional discovery bonuses. When an agent discovers aggressive yet valid high-frequency arbitrage strategies that exploit these properties, its realized total reward naturally exceeds the validator's conservative estimate, leading to normalized accuracy values above 100%. This behavior reflects the conservativeness of the heuristic upper bound rather than any bug or reward leakage in the environment.

## C.3 AGENT IMPLEMENATATION

**Agent**

```python
class ReActAgent(BaseAgent):
    """
    A solver agent that receives environment description and action space,
    then interacts with the environment step by step using LLM reasoning.
    """

    # ==== Core Attributes ====
    name: str = "solver"
    description: str = "A solver agent for solving the environment."
    current_action_space: str = ""
    past_actions: List[Dict[str, Any]] = []

    # ==== Core Functions ====

    def parse_action(self, resp: str) -> Dict[str, Any]:
        """Parse LLM response to structured action dict."""

    def _resolve_max_steps(self, env: SkinEnv, env_info: Dict) -> int:
        """Resolve maximum step limit with override precedence."""

    def _get_recent_actions(self):
        """Return recent action history as readable string."""

    # ==== Step Function, You Don't need to modify this =====
    async def step(self, agent_obs: Dict) -> Tuple[Dict, str]:
        act_prompt = AGENT_ACT_PROMPT.format(
            env_instruction=LEARNED_INSTRUCTION_PROMPT,
            action_space=self.current_action_space,
            obs=agent_obs,
            recent_actions=self._get_recent_actions(),
        )
        try:
            resp = await self.llm(act_prompt)
        except Exception as e:
            logger.error(f"LLM call failed: {e}")
            resp = None

        action = self.parse_action(resp)

        # First try to extract thinking_memory tag content
        if resp:
            thinking_memory_content = parse_xml_content(resp, "thinking_memory")
            if thinking_memory_content.get("thinking_memory"):
                thought = thinking_memory_content["thinking_memory"]
            # Fallback to original extraction logic if no thinking_memory tag found
            elif '```json' in resp:
                thought = resp.split('```json')[0].strip()
            elif '```' in resp:
                thought = resp.split('```')[0].strip()
            else:
                thought = resp.strip()
        else:
            thought = "No response from LLM"

        if thought:
            logger.agent_thinking(f"Agent Thought: {thought}")

        logger.agent_action(f"Agent Action: {action}")

        return action, thought

    # ===== Run Function, You Don't need to modify this =====

    async def run(self, env:SkinEnv, env_info:Dict):
        """
        env info:
        {
            "world_id": str,
            "agent_instruction": str,
            "action_space": str,
            "max_step": int,
        }
```

```
1566            """
1567            self.past_actions = []
1568
1569            world_id = env_info["world_id"]
                env.reset(mode="load", world_id=world_id)
1570
                self.current_action_space = env_info["action_space"]
1571
1572            # Resolve step limit with override-first precedence
                max_step = self._resolve_max_steps(env, env_info)
1573            cur_steps = 0
                cur_reward = 0
1574            events_count = {}
1575
1576            raw_obs = env.observe_semantic()
                agent_obs = env.render_skin(raw_obs)
1577            initial_observation = agent_obs
1578
                # Execute in Environments
1579            while cur_steps < max_step and not env.done():
                    logger.info(f"Environment Observation: \n{agent_obs}")
1580                action, thought = await self.step(agent_obs)
1581                _, reward, done, info = env.step(action)
                    # Record action along with last action result for better context
1582                self.past_actions.append({
1583                    "action": action,
                        "thought": thought,
1584                    "observation": agent_obs,
                        "result": info.get("last_action_result"),
1585                    "events": info.get("events", []),
                        "reward": reward,
1586                    "parse_error": (action or {}).get("_parse_error"),
1587                })
                    cur_reward += reward
1588                agent_obs = info["skinned"]
                    for e in info.get("events", []):
1589                    events_count[e] = events_count.get(e, 0) + 1
                    cur_steps += 1
1590                if done:
                        break
1591
1592            return {
                    "total_reward": cur_reward,
1593                "events_count": events_count,
                    "step": cur_steps,
1594                "initial_observation": initial_observation,
                }
1595
1596
1597
1598
1599
```

**Basic Prompt**

```
1602    AGENT_ACT_PROMPT = """
1603    ==== Environment Instruction ====
        {env_instruction}
1604
        ==== Action Space ====
1605    {action_space}
1606
        ==== Output Format ====
1607
        ==== thinking output format ====
1608    Before outputting an action, you should think step by step, and write any necessary
        reasoning (such as environment rules or information relevant for future actions) inside
1609    the <thinking_memory></thinking_memory> tag.
1610
        ==== Action Output Format ====
1611    When you output the action,
        you should output the action name and parameters in the format python dict can parse,
1612    and wrapped it in <action></action> tag, and only one action.
        Such as,
1613    <action>
        {{
1614        "action": "",
            "params": {{
1615            "<param_name>": "<param_value>"
            }}
1616    }}
```

```
</action>

The thinking and action should be outputted separately:
- First, write your reasoning inside <thinking_memory></thinking_memory> tag
- Then, output the action inside <action></action> tag, and the action content should can
be parsed by python dict.

==== Past Actions ====
Your recent actions are:
{recent_actions}

==== Now, your observation is:====
{obs}
"""

LEARNED_INSTRUCTION_PROMPT = None
```

# D    LEARNING DETAILS

## D.1    6-ENV SUBSET FROM AUTOENV-36

We select these six environments from AUTOENV-36 to form a compact subset that still covers the key capability axes in Table A7, while also spanning a broad range of difficulty levels and reward structures. The subset is designed to jointly probe navigation (Nav), partial observability and memory (POM), counterintuitive or inverted rules (Inv.), control and resource management (Ctrl/Res), pattern and symbolic reasoning (Patt), and long horizon planning (Plan), under both dense and very sparse rewards and with varying degrees of semantic deception.

InterDimension (1-ID) mainly covers Ctrl/Res, Patt and Plan, using high dimensional resources and exchange rates to evaluate multi objective control and pattern modeling under noisy observations, with cumulative profit rewards that require stable long horizon policies. Backwards Valley Farm (19-AS) covers Nav, POM, Inv. and Ctrl/Res, combining a partially observable map with counterintuitive farming rules to evaluate causal reconstruction and planning under misleading everyday intuitions, with relatively sparse rewards that are easy to hurt with wrong habits. Deceptive Grid World (21-WN) covers Nav, POM, Inv. and Patt, and uses systematically deceptive symbols to test whether an agent can rebuild the true world model and plan paths from local views under very sparse rewards. Chaos Slide Puzzle (24-MM) covers Patt and Plan, acting as a pure symbolic sliding puzzle where only the single chaotic goal configuration yields reward, which evaluates precise sequence planning and combinatorial search without intermediate shaping. Mismatched Memory Game (26-TA) covers POM, Patt and Plan, evaluating working memory and hidden rule discovery in randomized card layouts, where reward comes from repeated correct matches within a single episode. Alien Colony Management (33-AD) covers Ctrl/Res, Inv. and Plan, using semantically misleading resources and buildings to evaluate complex system modeling and long term stable control, with multi variable cumulative rewards that are highly sensitive to poor decisions.

Taken together, these six environments jointly test Nav, POM, Inv., Ctrl/Res, Patt and Plan, and provide complementary reward densities and difficulty profiles. This makes them a representative six environment subset of AUTOENV-36 for analyzing cross environment learning behavior in more detail.

## D.2    ALGORITHM OF AGENT LEARNING

## D.3    LEARNING FORMULATION OF FOUR TYPICAL AGENT LEARNING FRAMEWORK.

Within our Selection–Optimization–Evaluation (S/O/E) framework, existing agentic-learning methods can be interpreted as operating on different types of candidates with different selection rules and optimization signals.

**AFlow.** AFlow treats the agentic workflow (the full workflow program) as the candidate. Selection uses a mixed "best-and-random" strategy: it keeps high-performing workflows while also sampling unexplored variants. Optimization is implicit and LLM-driven: the language model rewrites or

---

**Algorithm 2** Component-Centric Agent Learning Framework

---

**Require:** Environment $\mathcal{E}$, initial candidate $c_0$, selection function $\mathcal{F}_s$, optimization function $\mathcal{F}_o$, evaluation function $\mathcal{F}_e$, max iterations $T$
**Ensure:** Candidate pool $P$
1: $P \leftarrow \{c_0\}$
2: **for** $t = 1$ to $T$ **do**
3:     $C_{\text{sel}} \leftarrow \mathcal{F}_s(P)$                                          ▷ Selection step
4:     $C_{\text{new}} \leftarrow \mathcal{F}_o(C_{\text{sel}})$                    ▷ Optimization step, update target components
5:     **for** $c \in C_{\text{new}}$ **do**
6:         $(\tau_c, m_c) \leftarrow \mathcal{F}_e(c, \mathcal{E})$     ▷ Evaluation step, run in environment and compute metrics
7:         $c.\text{trajectory} \leftarrow \tau_c$
8:         $c.\text{metrics} \leftarrow m_c$
9:     **end for**
10:    $P \leftarrow P \cup C_{\text{new}}$
11: **end for**
12: **return** $P$

---

extends workflows based on execution traces without an explicit numeric reward. Evaluation is performed on downstream benchmarks, where task-level scores are used to compare workflows.

**SPO.** SPO operates directly on prompts as candidates. Selection uses a current-best rule, keeping the best-performing prompt and generating local variants around it. Optimization is again LLM-based, guided by feedback from a judge model but without an explicit external reward function. Evaluation relies on LLM-as-a-judge and pairwise comparison of outcomes, which induces a preference ordering over prompts.

**GEPA.** GEPA also takes prompts as candidates, but frames learning as multi-objective optimization. Selection keeps candidates on the Pareto front with respect to several metrics (e.g., accuracy, robustness, complexity). Optimization combines explicit reflection signals—LLM-generated explanations of errors and improvement directions—with prompt mutations to generate new candidates. Evaluation is iterative on benchmarks, repeatedly scoring prompts on multiple objectives to update the Pareto front.

**Darwin Gödel Machine (DGM).** DGM treats the full agent code (including tools, controller, and self-modification logic) as the candidate. Selection chooses code variants based on both their empirical performance and their position in the "lineage" (parent–child relationships). Optimization is driven by attribution on error: the system analyzes failures, attributes them to specific parts of the code, and rewrites those components. Evaluation is carried out on benchmark suites, where task performance provides the main signal for comparing and selecting code variants. Within our Selection–Optimization–Evaluation (S/O/E) framework, existing agentic-learning methods can be interpreted as operating on different types of candidates with different selection rules and optimization signals.

**AFlow.** AFlow treats the agentic workflow (the full workflow program) as the candidate. Selection uses a mixed "best-and-random" strategy: it keeps high-performing workflows while also sampling unexplored variants. Optimization is implicit and LLM-driven: the language model rewrites or extends workflows based on execution traces without an explicit numeric reward. Evaluation is performed on downstream benchmarks, where task-level scores are used to compare workflows.

**SPO.** SPO operates directly on prompts as candidates. Selection uses a current-best rule, keeping the best-performing prompt and generating local variants around it. Optimization is again LLM-based, guided by feedback from a judge model but without an explicit external reward function. Evaluation relies on LLM-as-a-judge and pairwise comparison of outcomes, which induces a preference ordering over prompts.

**GEPA.** GEPA also takes prompts as candidates, but frames learning as multi-objective optimization. Selection keeps candidates on the Pareto front with respect to several metrics (e.g., accuracy, robustness, complexity). Optimization combines explicit reflection signals—LLM-generated explanations of errors and improvement directions—with prompt mutations to generate new candidates.

Evaluation is iterative on benchmarks, repeatedly scoring prompts on multiple objectives to update the Pareto front.

**Darwin Gödel Machine (DGM).** DGM treats the full agent code (including tools, controller, and self-modification logic) as the candidate. Selection chooses code variants based on both their empirical performance and their position in the "lineage" (parent–child relationships). Optimization is driven by attribution on error: the system analyzes failures, attributes them to specific parts of the code, and rewrites those components. Evaluation is carried out on benchmark suites, where task performance provides the main signal for comparing and selecting code variants.

### D.4 LEARNING PROMPT

---

**Signal Prompt**

```
"""
Signal prompt templates for trajectory-driven analysis (plain text output).

These prompts ask for concise normal text. The raw text becomes the signal content
ingested by the optimization stage. Do NOT require XML or YAML in the output.
"""

# 1) Dynamics-focused analysis
# Goal: infer environment rules, rewards, transitions, hazards from trajectories.
DYNAMICS_OPTIMIZATION_PROMPT = """
You are an expert at reverse-engineering environment dynamics from agent trajectories.

Input trajectories (human-readable):
{trajectories}

Optional current component (prompt or agent code excerpt):
{component_content}

Write a concise analysis (plain text) that covers:
- Environment: key state variables, observations, and action space the agent appears to
have.
- Transitions: common preconditions → effects; typical progress vs. dead-ends; termination
cues.
- Rewards: which actions/events correlate with reward changes; signs of sparse/dense
reward.
- Failures: frequent mistakes and likely causes, with brief evidence from the
trajectories.
- Strategies: practical heuristics/rules to increase reward and reduce mistakes.
- Uncertainties: what remains unclear and what evidence would disambiguate it.
- Confidence: your overall confidence (0.0{1.0}).
"""

# 2) Instruction-focused analysis
# Goal: diagnose why the agent underperforms and propose instruction changes.
INSTRUCTION_OPTIMIZATION_PROMPT = """
You evaluate agent trajectories to improve the agent's instruction/policy prompt.

Input trajectories (human-readable):
{trajectories}

Optional current instruction/code excerpt:
{component_content}

Write a concise diagnosis and proposal (plain text) that covers:
- Diagnosis: concrete failure patterns
(perception, action choice, planning, termination misuse, etc.).
- Principles: short, general rules the agent should follow (imperative and checkable).
- Step Guidelines: when-then style rules for common situations.
- Guardrails: behaviors the agent must avoid, with conditions.
- Mini Examples (optional): tiny templates that illustrate correct handling.
- Measurement: how success should be measured and expected direction of change.
- Confidence: your overall confidence (0.0{1.0}).
"""
```

---

## D.5 LEARNING CASE

### D.5.1 LEARNED PROMPT

---

**Agent Instruction**

---

```
You are in a 3x3 sliding tile puzzle environment with an inverted objective.
Your task is to arrange the numbered tiles (1-8) and blank space into a specific chaotic
pattern within exactly 30 moves.
The target pattern is: [[2,7,5],[1,0,8],[4,6,3]] where 0 represents the blank space.
You must avoid creating the conventional ordered pattern [[1,2,3],[4,5,6],[7,8,0]] as this
causes immediate failure. You can slide tiles by moving the blank space in four directions
(up, down, left, right),
but only if the movement is within the 3x3 grid boundaries.
Each move consumes one step regardless of whether it's legal or illegal,
so plan your moves carefully to reach the chaos pattern before running out of steps.
```

---

---

**Learned Agent Instruction**

---

```
You are an agent that plays a 3 × 3 sliding{tile puzzle with a twist.

Environment
• Observation: a 3 × 3 list of lists whose numbers 1–8 are tiles and 0 is the blank;
you may also be shown the remaining-move counter (starts at 30).
• Action space: { "up", "down", "left", "right" } { interpreted as sliding the
blank one cell in that direction.
• Transition rules
  { Legal: the blank swaps with its neighbour in the chosen direction and the move counter
  decrements by 1.
  { Illegal: the board stays the same but the move counter still decrements by 1.
• Termination
  { Success: the board equals [[2, 7, 5], [1, 0, 8], [4, 6, 3]] at or before the 30-move
  limit.
  { Instant failure: the board ever equals the classic ordered pattern [[1, 2, 3],
  [4, 5, 6], [7, 8, 0]].
  { Ordinary failure: the move counter reaches 0 without success.

Constraints & advice
1. Never output more than one action per turn.
2. Each message must have two parts in the following order:
  a) Your private reasoning wrapped in a single pair of <thinking_memory>
  </thinking_memory> tags.
  b) Exactly one JSON object containing the chosen action, with this template:
    {
        "action": "<one_of_up_down_left_right>",
        "params": {}
    }
3. Do not output anything else|no code fences, no commentary.
4. Aim to solve the puzzle within <= 30 moves while avoiding the canonical solved pattern
and illegal moves.
5. Once success or any failure condition is reached, stop acting.

Follow this format strictly so your output can be parsed programmatically.
```

---

### D.5.2 LEARNED AGENT

---

**Learned Agent**

---

```python
async def step(self, agent_obs: Dict) -> Tuple[Dict, str]:
        """Enhanced step function with better context and error handling"""
        current_step = len(self.past_actions)
        max_step = 30  # Default, will be updated in run()

        # Generate memory summary
        memory_summary = self._get_memory_summary()

        act_prompt = AGENT_ACT_PROMPT.format(
            env_instruction=LEARNED_INSTRUCTION_PROMPT,
            action_space=self.current_action_space,
```

```
                obs=agent_obs,
                recent_actions=self._get_recent_actions(),
                current_step=current_step,
                max_step=max_step,
                exploration_strategy=self.exploration_strategy,
                memory_summary=memory_summary
            )

            try:
                resp = await self.llm(act_prompt, max_tokens=1000)
            except Exception as e:
                logger.error(f"LLM call failed: {e}")
                # Fallback to simple exploration
                return {"action": "MoveNorth", "params": {}},
                "LLM failed, using fallback action"

            action = self.parse_action(resp)

            # Extract thinking
            thought = "No thought provided"
            if resp:
                thinking_content = parse_xml_content(resp, "thinking_memory")
                if thinking_content.get("thinking_memory"):
                    thought = thinking_content["thinking_memory"]
                else:
                    # Fallback extraction
                    if '<thinking_memory>' in resp:
                        start = resp.find('<thinking_memory>') + len('<thinking_memory>')
                        end = resp.find('</thinking_memory>')
                        if end > start:
                            thought = resp[start:end].strip()

            logger.agent_thinking(f"Agent Thought: {thought}")
            logger.agent_action(f"Agent Action: {action}")

            return action, thought
```

### D.5.3 LEARNING TRAJECTORIES IN ENV_19

To better illustrate how different components are updated during learning, we analyze three representative trajectories on the same environment `19_AgriculturalSimulation` (*Backwards Valley Farm*). The three settings differ in which components are treated as learnable:

| Case | Configuration | Iters | Prompt-only | Agent-code-only | Both |
|------|---------------|-------|-------------|------------------|------|
| 1 | instruction_prompt | 10 | **10** | 0 | 0 |
| 2 | instruction_agent | 10 | 0 | 0 | **10** |
| 3 | dynamics_agent | 10 | 0 | 0 | **10** |

**Case 1: `instruction_prompt` (prompt-only learning).** In this setting, the only learnable component is the environment instruction prompt, while the ReAct agent code is frozen. Across all 10 iterations, the learner only edits the instruction prompt. The initial prompt is a generic description that encourages exploration but lacks clear priorities. Over iterations, the prompt is refined into a highly structured policy: always observe before acting, maintain an explicit priority ordering over actions (e.g., harvesting mature crops > collecting animal products > feeding > planting > trading), and execute exactly one highest-value action per step. On Backwards Valley Farm, which combines semantically counter-intuitive rules with dense rewards, this structured instruction alone significantly improves performance (accuracy increases roughly from $0.25$ to $0.47$) without changing the agent implementation.

**Case 2: `instruction_agent` (instruction-implementation as a joint component).** Here, the optimized component is the "instructional agent": a combination of a natural-language instruction describing how the agent should behave and the corresponding agent implementation. In practice, all 10 iterations modify both the instruction prompt and the agent code. Prompt edits mainly refine how the agent should record trajectories, handle exceptions, and interact with multiple worlds. Code edits focus on making the implementation robust—more reliable environment resets, safer trajectory logging, and graceful failure handling instead of aborting the entire run. This reflects our design

choice that, when the component is defined as an *instructional agent*, both the textual description and its faithful implementation are part of the same learnable unit.

**Case 3: `dynamics_agent` (strategy-structure learning).** In the `dynamics_agent` setting, the agent code is explicitly allowed to implement internal strategy analysis and simple dynamics modeling. Again, all 10 iterations update both the agent code and its associated prompts. Compared to a standard ReAct agent that reacts myopically to the current observation, the improved agent introduces a structured decision process: it maintains statistics over recent actions and rewards, summarizes these statistics into high-level "strategy insights" via a dedicated analysis prompt, and then conditions action selection on both the current state and these insights. On Backwards Valley Farm, this leads to clear performance gains (e.g., from an initial accuracy of about $0.28$ to $0.31$ with roughly half the cost), as the agent can exploit the stable but counter-intuitive reward structure by explicitly tracking which actions have been consistently beneficial.

**Summary of component behavior.** These trajectories illustrate how our component definitions translate into concrete update patterns. When prompts are treated as standalone components (e.g., `instruction_prompt`), we freeze the agent code and attribute all changes to the evolution of the instruction prompt. When the component is defined as an *agent* (e.g., `instruction_agent`, `dynamics_agent`), the search space includes both the Python implementation and tightly coupled prompt templates, allowing the agent to become more robust (Case 2) or to acquire richer internal strategy structures (Case 3).

# E  ADDTIONAL EXPERIMENTS

## E.1  DETAILED PERFORMANCE ON AUTOENV-36 LEARNING EXPERIMENT

Table A9: Learning method performance on all 36 environments using Gemini-2.5-Flash. We test 4 methods under Best Selection across different environment types. Upper bound selects the best method per environment.

| Method | Reward | | Observation | | Semantic | | Acc |
|---|---|---|---|---|---|---|---|
| | Binary | Accum. | Full | Partial | Aligned | Inverse | |
| IO | 43.89 | 34.93 | 46.84 | 34.10 | 38.95 | 41.02 | 39.41 |
| Dynamics + Prompt | 43.43 | 41.38 | 53.04 | 34.80 | 42.00 | 43.80 | 42.40 |
| Dynamics + Agent | 41.62 | 39.10 | 52.93 | 31.38 | 40.40 | 40.21 | 40.36 |
| Instruction + Prompt | 44.03 | 39.09 | 50.66 | 35.06 | 41.00 | 43.54 | 41.56 |
| Instruction + Agent | 38.89 | 36.63 | 45.79 | 32.02 | 37.71 | 37.93 | 37.76 |
| UpperBound | 50.32 | 45.18 | 58.52 | 40.06 | 47.48 | 48.71 | 47.75 |

## E.2  SFT SETUP

We include an in-domain supervised fine-tuning (SFT) baseline on the same six benchmarks used in our learning experiments. The base model is Qwen-2.5-7B-Instruct, and the goal is to provide a per-environment specialized training-based method for comparison with the inference-time learning strategies, rather than to claim cross-environment generalization.

For each of the six environments, we use the environment validator to generate new levels and run ReAct-style agents backed by DeepSeek-V3.1, GPT-5, O3, and Claude-4-Sonnet. Each agent action is treated as one training example, where the input prompt contains the system prompt, observation and interaction history up to step $t$, and the response is the next action at step $t+1$. We ensure 800 examples per environment and split them into 80% training and 20% validation, independently for each environment.

We train a standard SFT objective that maximizes the likelihood of the response given the prompt. Training uses a maximum sequence length of 4096 tokens, global batch size 16 (micro-batch size 4 per GPU), bf16 mixed precision, and 8 epochs on 4 GPUs with an FSDP-based trainer. Under this setup, SFT should be interpreted as an in-domain, per-environment specialization baseline for these six benchmarks.

### E.3 SKIN INVERSE EXPERIMENT

To better understand why environments with inverse semantics sometimes yield higher scores than aligned ones, we run a "Skin Inverse" ablation. The key idea is to *invert only the Skin / observation layer*—i.e., the symbols and values shown to the agent—while keeping the underlying BaseEnv dynamics and reward function unchanged. This allows us to probe whether agents truly reason about semantic inversions, or whether higher scores are mostly due to easier structural difficulty.

We instantiate this ablation on three representative benchmarks, covering different types of semantic inversions: numerical label inversion (Benchmark 14), symbol-level inversion (Benchmark 20), and pixel-value inversion (Benchmark 31). Unless otherwise noted, we report results with Gemini-2.5-Flash as the execution model.

**Benchmark 14: FieldDetection (electromagnetic anomaly mapping).** In the original environment, the agent receives a local $3\times3$ patch of field intensities and must locate a fragile "critical" node. In the inverted Skin version, we modify env_obs.py to *invert the displayed field strength* ($3\rightarrow0$, $2\rightarrow1$, $1\rightarrow2$, $0\rightarrow3$), and the textual legend claims that "0 = Critical, 3 = Stable" while the true goal remains the physical intensity $0$. We also explicitly warn the agent that "the environment may contain semantic inversions".

Gemini-2.5-Flash frequently produces internal reasoning indicating that it suspects such an inversion (e.g., hypothesizing that the true target may be zones labelled as "Critical (0)" rather than "Stable (3)"), and it does navigate towards these regions. However, execution remains unstable: over three runs the average normalized score is $26.67\%$ (two episodes reaching partial reward around $0.4$, one failing with zero reward). This suggests that the model can conceptually reason about the inversion, but has difficulty turning this into consistently successful control.

**Benchmark 20: GridNavigation (simplified inverted grid world).** We construct a simplified grid navigation environment where the agent must reach a treasure within $40$ steps. In the Skin-inverted variant, we swap the rendering of empty cells and walls in env_main.py: Empty tiles are drawn as "#" and Wall tiles as ".". The legend and instructions still claim that ". = Path" and "# = Obstacle", but in reality the agent can only move on "#" and collides on ".".

Initially, Gemini interprets the symbols literally and finds itself surrounded by apparent obstacles. After successfully issuing a move action (e.g., MOVE_NORTH) through a ring of "#" tiles, it explicitly reasons that this contradicts the stated legend and infers that "#" must actually be traversable and "." non-traversable. In other words, the agent *discovers the symbol inversion through interaction*. Despite this correct inference, its exploration strategy remains inefficient: in our runs it repeatedly follows simple directions and fails to systematically search for the treasure, leading to an average normalized score of $0.0\%$ within the 40-step budget.

**Benchmark 31: PatternCompletion (masked pixel art completion).** In the pixel-art completion task, the agent moves a cursor on a $10\times10$ canvas and writes discrete color values to fill masked pixels. In the Skin-inverted version, we invert the displayed color values in the renderer (color_shown = 15 - color_true) and inform the agent that "higher signal numbers typically indicate stronger features". Thus a region that visually appears as value $15$ actually corresponds to true color $0$, and the correct action is often to write color $0$ instead of $15$.

Here, Gemini fails to detect the inversion. Given neighbors such as $(15, 15, 11)$, it consistently reasons that the majority value $15$ should be written at the masked location, and issues actions like WriteColor15. The environment returns an incorrect_action event and zero reward, but the agent continues to apply the same heuristic for many steps, without abstracting the global "$15 \leftrightarrow 0$" flip. Across episodes, it only rarely fills a pixel correctly by chance, yielding an average normalized score of $8.62\%$. Qualitatively, the agent shows almost no ability to learn the inversion from repeated fine-grained feedback.

**Summary.** Table A10 summarizes these three Skin-inverted benchmarks. Overall, we observe that: (i) with an explicit hint about possible semantic inversions, the agent can often *verbalize* or locally infer that labels may be flipped (Benchmarks 14 and 20); (ii) however, turning such meta-level hypotheses into stable, reward-maximizing policies is still difficult; and (iii) in settings where

the inversion must be inferred purely from sparse reward or `incorrect_action` signals (Benchmark 31), the agent largely fails to adapt. These results support our claim that higher scores on inverse-semantic environments do not imply robust inversion handling; instead, they are largely driven by structural difficulty, and current agents still exhibit a sizable gap between language-level reasoning about inversions and behavioral adaptation.

Table A10: Summary of Skin Inverse experiments on three benchmarks with Gemini-2.5-Flash as execution model.

| Bench. | Inversion type | Understood inversion? | Avg. score |
|---|---|---|---|
| 14 (FieldDetection) | Numeric label inversion | Often (concept-level) | 26.67% |
| 20 (GridNavigation) | Symbol / tile semantics inversion | Yes (via interaction) | 0.00% |
| 31 (PatternCompletion) | Pixel value display inversion | No (fails to adapt) | 8.62% |

## E.4 MULTIMODAL SKIN GENERATION EXPERIMENT

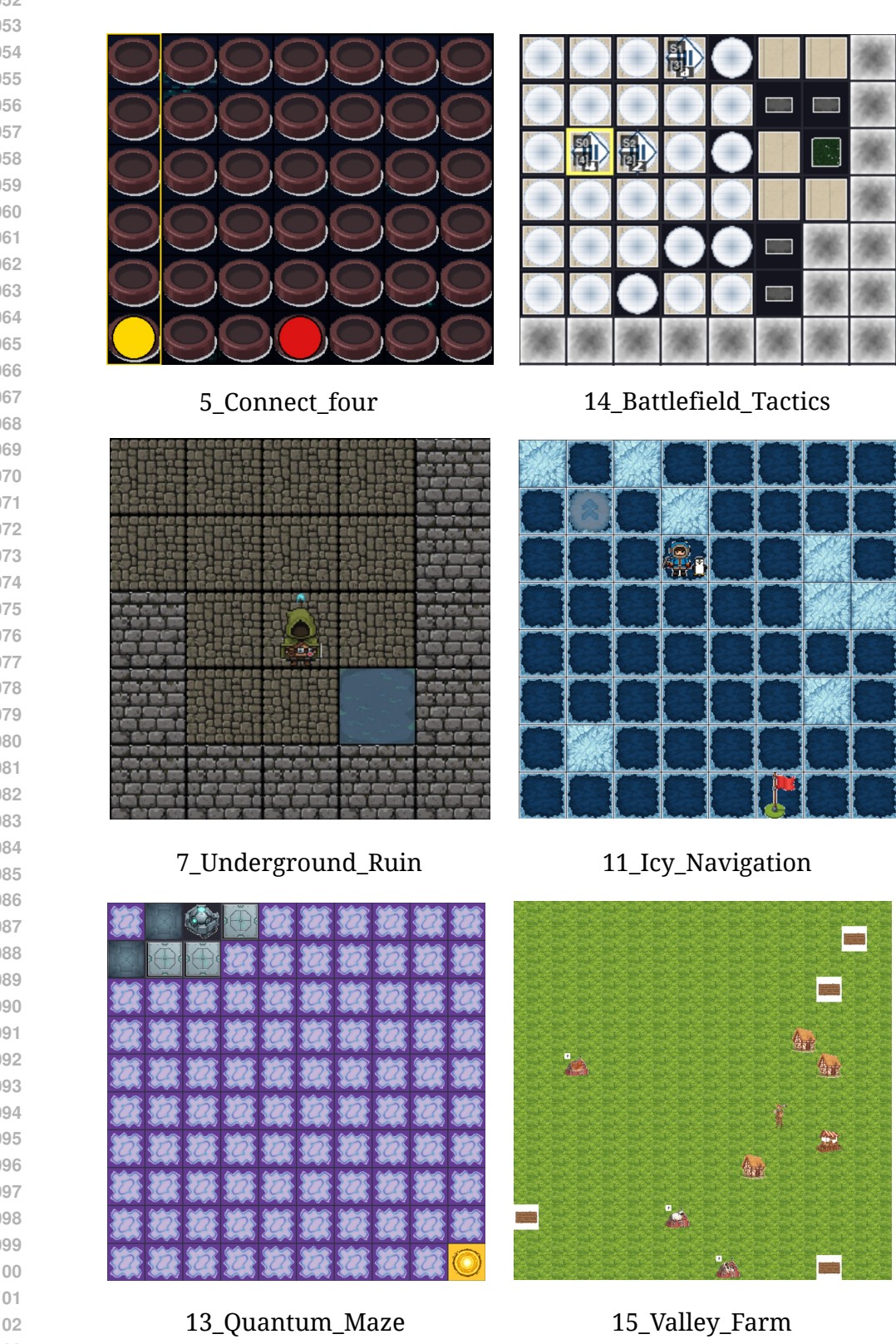

Figure A1: Generated multimodal skin for partial environments in AUTOENV-36.

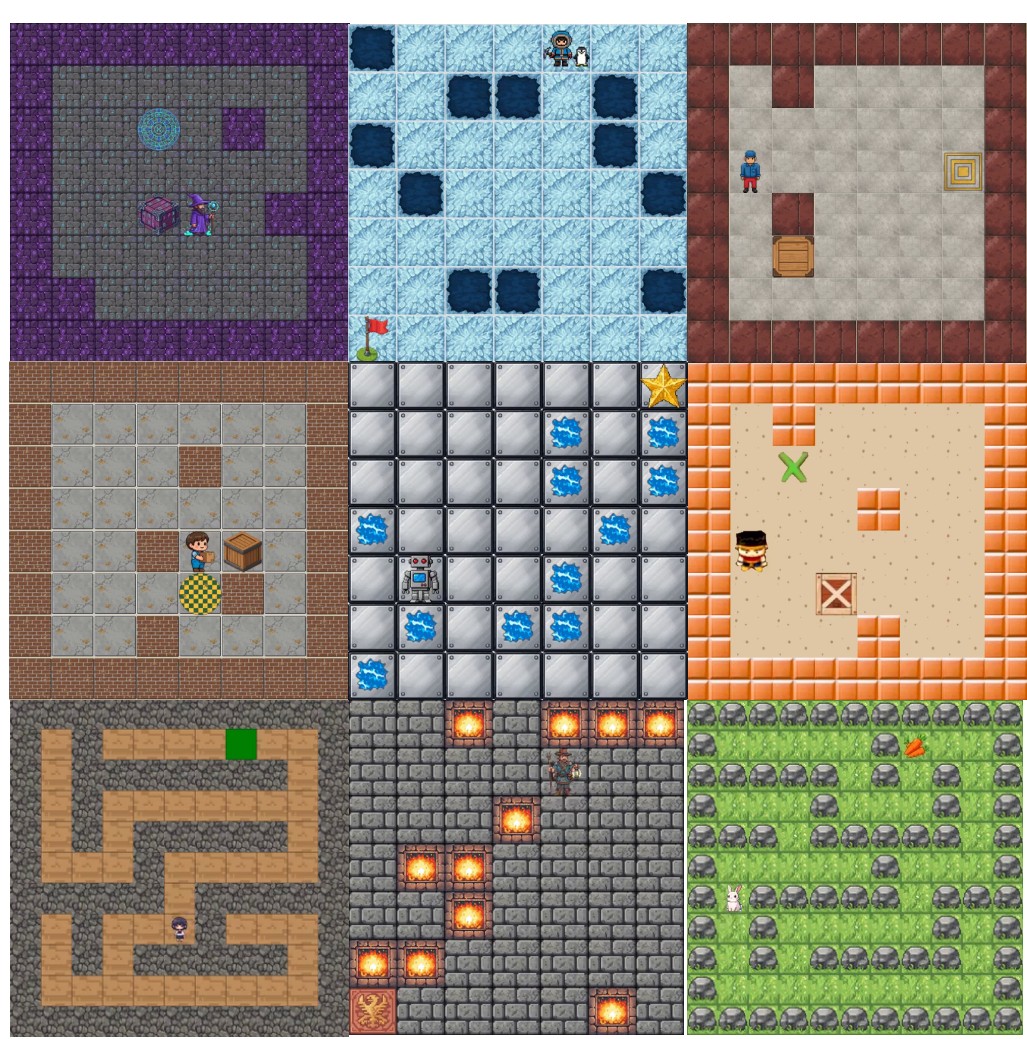

Figure A2: Generated multimodal skins for the same base environment, showing how rules can be decoupled from observations.

