# OpenReview forum: "Automating Environments For Measuring Agentic Learning"
_ICLR.cc/2026/Conference — Submitted to ICLR 2026_

### Official Review · Reviewer_FGMv · 2025-10-28

**Soundness:** 3
**Presentation:** 2
**Contribution:** 2
**Rating:** 4
**Confidence:** 4

**Summary:**

This paper introduces AUTOENV framework to automatically generate different envrioments for evaluating and training agentic language models. The author states that current agents lack the ability to generalize across different rule systems due to the limit and human-designed environments with fixed policies. Thus, the proposed AUTOENV decompose environments intro BaseEnv, ObsEnv, and SkinEnv abstract layers and provide variation in reward structures, state dynamics, and partial observability. This paper also formalizes agentic learning as modular components including selection strategies, optimiztion signals, and target components. Extensive experiments shows the effectiveness of this framework.

**Strengths:**

1. Originality: The proposed AUTOENV automantes the envrionment generation for agentic learning. While it builds upon exisiting ideas from benchmark construction and meta-learning, the automated environment creating with structured learning strategy is well-motivated.
2. Quality: The technical execution is solid. The three-layer abstraction is clearly presented and the working pipeline capable of producing validated and executable environments is demonstrated.
3. Clarity: The manuscript is well orgnanized and systematically written.
4. Significance: This work addresses an important problem of the lack of scalable and diverse environments for evaluation.By automating envrionment creation and framing agentic learning as composable components, this paper shows potential contributions to future research in generalist and adaptive AI.

**Weaknesses:**

While this paper shows credible contribution, several weaknesses limit its overall impact.
1. Despite that the automated envrionment creating with structrued learning strategy is well-motivated, the conceptual advancement lies in combining the automated benchmark generation and meta-learning components rather than introducing fundamentally new solutions. Therefore, the contribution is well-motivated yet incremental.
2. Although this paper reports that the AUTOENV-36 contains 36 validated heterogeneous envrionments, the analysis is largely based on decriptive rather than quantitative. It is unclear how distinct these envrionment are in terms of rule distribution, learning dynamics, and transfer difficulty.
3. The reported 74.7% consistency validation rate may suggest that nearly 25% generated environments may exhibit unstable or inconsistent reward and rule behaviors, which may raise concerns about the robustness.
4. The experiments exclusively evaluate LLM agents without comparison to non-LLM or classical RL baselines. This makes it difficult to asses whether the claimed improvments are due to the adaptive learning or simple model capacity differences.
5. Besides, the experiments also evaluate closed-sourced LLMs without open-sourced LLMs after finetuning. It is unclear whether the introduced complexity from the integrating of diverse envrionments still challenges the current fine-tuned LLMs, which weakens the influence of the contributions of this work.

**Questions:**

Based on the previous description, I have some questions listed below.
1. Could the author provide some quantitive measures of diversity among the generated envrionments beyond categorical features. How do AUTOENV ensure that the generated envrionments differ meaningfully  in their underlying mechanics instead of just parameter variations?
2. This paper reports a 74.7% consistency validation rate. Can you elaborate on what kinds of inconsistencies occur during the rest 25%?
3. Due to the fact that the envrioments are generated in closed loops by large language models, are there bottlnecks such as model inference time or code validation loops that might limit the scalability of larger envrionment amounts needed?
4. How does AUTOENV-36 compare quantitatively to other benchmark collections such as GG-Bench or OSWorld in terms of diversity, difficulty, and adaptability?
5. Are there safeguards to prevent overfitting or environment leakage during generation and validation such as environments unintentionally reusing the same rule patterns?
6. The paper briefly mentions possible future directions such as multimodal and embodied scenarios and the chosen primary area is applications to robotics. According to the demonstrations in the Appendix, the envrionments genreated are not complex. Could the authors outline what modifications would be needed to extend AUTOENV beyond text-based settings, for example, to 3D embodied or visual environments?
7. Is the AUTOENV able to generate sceanarios with more envrionment non-stationarities, such as competitive envrioment with diversity opponent strategies?

---

> ### Author Response · Authors · 2025-11-27
> **Response to Reviewer FGMv 1/3**
>
> We thank the reviewer for the careful and constructive feedback. We are glad that you found the framework well-motivated and technically solid, and we have further strengthened the revised version along both experimental and conceptual dimensions. Below, we first address the experimental concerns around diversity, robustness, baselines, and scalability, and then clarify the conceptual contribution and potential extensions.
>
> ## Response to W2/Q1/Q4/Q5: Environment diversity & comparison
>
> ### (W2 & Q1) Diversity and distinct mechanics.
>
> We agree that the initial version relied too much on descriptive arguments. In the revision, we make the diversity of AUTOENV-36 more explicit in two parts:
>
> - In Section 3.3 and Appendix A7, we provide quantitative statistics over three core feature families—reward rules, observation structures, and semantic alignment—together with a capability-level taxonomy for all 36 environments (e.g., planning depth, partial observability, memory demands, exploration difficulty, combinatorial reasoning).
> - Beyond categorical labels, we describe how different environments instantiate distinct underlying mechanics: for example, grid navigation with traps, resource-management farms, and multi-stage procedural tasks. We also explain how their generators and validators induce different learning dynamics and transfer difficulty.
>
> This makes clear that AUTOENV-36 is not produced by simple parameter variations, but by qualitatively different rule systems synthesized from distinct themes.
>
> ### (Q4) Comparison to other benchmarks (GG-Bench, OSWorld).
> Our setting is closest in spirit to GG-Bench, which also aims at automatically constructing environments, but focuses specifically on two-player competitive games. AUTOENV aims broader: our three-layer abstraction is designed to build a diverse collection of agentic environments, not a single game family. Each environment has its own generator and validator, targeting different rule structures and skills.
>
> Compared to OSWorld, which targets a fixed collection of complex, hand-designed tasks, AUTOENV emphasizes scalability of environment instances. Because each of the 36 environments in AUTOENV-36 comes with reusable generators and validators, we can derive large numbers of levels per environment, yielding a total number of instances that already exceeds many existing collections. We have added a discussion of this trade-off—automatic breadth vs. hand-crafted depth—in Section 3.3.
>
> ### (Q5) Preventing leakage and “reskinned” environments.
>
> Each environment is generated from a distinct high-level theme description. During theme design, we explicitly prompt the model to produce semantically and structurally different themes and discard those that are too close to existing ones. In the curation step for AUTOENV-36, coding agents inspect the DSL and generated code to detect redundant rule patterns, and we retain environments with genuinely different mechanics (including some with lower success rates but distinct rules).
>
> Together, we hope these changes and clarifications can address W2, Q1, Q4, and Q5 by making the diversity and distinctiveness of AUTOENV-36 more concrete and by positioning it relative to existing benchmarks.

---

> ### Author Response · Authors · 2025-11-27
> **Response to Reviewer FGMv 2/3**
>
> ## Response to W3/Q2: Environment generation robustness
>
> We appreciate the concern about robustness raised by the reported 74.7% consistency rate. This number, however, should be interpreted as a pipeline filtering statistic, not as a measure of instability in the final benchmark.
>
> In Section 3.2 (Line 209 in the revised version), we now clearly describe the three-stage verification process (execution, level generation, and what we now call reliability instead of “consistency”). The reported 74.7% refers to the fraction of environments that pass the reliability stage among those that already passed execution and level generation in our 100-theme study.
>
> For the remaining ~25% that fail the reliability stage, the typical issues include:
>
> - rare edge cases where the reward function behaves unexpectedly;
>
> - mismatches between the intended rule description and the validator’s behavior;
>
> - cases where the solver and validator disagree on the maximal achievable reward.
>
> Crucially, all such environments are discarded and do not enter AUTOENV-36. Thus, the ~25% failures are used purely as a filtering criterion to improve robustness, not as part of the benchmark itself. We have clarified this in the revised text to avoid the misunderstanding that 25% of the benchmark environments are inconsistent.
>
>
> ## Response to W4–W5: Baselines and fine-tuned open-source models
>
> We thank the reviewer for encouraging us to broaden the baselines.
>
> Our work is specifically targeted at LLM-based agents, and our main comparisons are within the same model with and without learning, so that improvements can be attributed to the learning strategy rather than to raw model capacity. For this reason, we do not include non-LLM or classical RL baselines in this paper, as they require different interfaces and would shift the focus away from the agentic LLM setting we aim to study.
>
> That said, we have extended the empirical coverage in the revised version in response to your suggestions:
>
> - In Table 4 (Section 5.3; Appendix E.2), we add a new experiment with an open-source model (Qwen-2.5-7B-Instruct) and a simple SFT-based learner integrated into our framework. This directly addresses the question of whether fine-tuned open-source LLMs still find AUTOENV-36 challenging.
>
> - Interestingly, under our component-centric learning formulation, the SFT learner is not uniformly superior: on some environments it is outperformed by training-free strategies. This supports our broader claim that no single learning method dominates, and that environment-specific or environment-adaptive strategies remain important even for fine-tuned models.
>
> ### Response to Q3: Environment scalability
>
> Regarding scalability, AUTOENV is designed so that each environment is constructed independently from its theme, using a bounded sequence of LLM calls and validation steps:
>
> - Given a suitable theme, the pipeline either (i) produces a fully validated environment (with BaseEnv, ObsEnv, SkinEnv, generator, and validator), or (ii) fails and rejects the theme after a bounded number of attempts. There is no global “closed loop” across environments that could deadlock or cause cascading failures.
>
> - Thanks to the three-layer abstraction and DSL, each call to the model produces bounded-length code and specifications; we never require the model to emit arbitrarily large monolithic programs. This makes the process stable with respect to context length and token limits.
>
> We have clarified this in the revision to emphasize that, while the pipeline is non-trivial, it is not fundamentally limited by code-loop instabilities and scales well to larger numbers of environments.

---

> ### Author Response · Authors · 2025-11-27
> **Response to Reviewer FGMv 3/3**
>
> ## Response to W1: Incremental contribution
> We respectfully disagree with the assessment that AUTOENV is merely an incremental combination of benchmark construction and meta-learning components. Our goal is to provide a unified framework that simultaneously addresses (i) the lack of scalable, heterogeneous environments, and (ii) the lack of a principled way to study cross-environment agent learning.
>
> Concretely, the contribution of the revised paper is fourfold:
>
> - Framework. We introduce a three-layer environment abstraction (BaseEnv / ObsEnv / SkinEnv) coupled with a DSL that factors reward structures, state dynamics, and observations. This enables not only low-cost generation, but also systematic semantic manipulations (e.g., aligned vs. inverse semantics, text-only vs. multimodal skins) under fixed underlying rules.
>
> - Dataset. On top of this framework, we construct AUTOENV-36, a curated set of 36 validated environments with heterogeneous rule distributions and 358+ levels, selected from 100 themes via a multi-stage generation and verification pipeline. In the revision, we further provide a capability-level taxonomy and feature analysis (Section 3.3; Appendix A7), making explicit what skills these environments probe.
>
> - Learning formulation. We formalize agentic learning as a component-centric process over four basic objects (candidates, components, trajectories, metrics) and three stages (selection, optimization, evaluation). In Appendix D.3 we show how several representative methods (SPO, GEPA, AFlow, DGM) are expressed within this formulation, so AUTOENV is not just a benchmark, but also a common testbed and language for existing learning approaches.
>
> - Experiments and insights. We scale both the model side (7 LLMs, including an open-source model with SFT in Table 4) and the environment side (from 6 to 36 environments), and define a Learning Upper Bound via an oracle that selects the best learning method per environment. The new experiments reveal several non-trivial insights:
>   - no single learning strategy dominates across heterogeneous environments;
>
>   - the gain of any fixed learning method decays rapidly as the environment set grows;
>
>   - even simple environment-adaptive selection recovers a significant fraction of the Learning Upper Bound, yet a clear cross-environment generalization gap remains.
>
> We believe this combination of a generative environment framework, a curated and analyzed dataset, a unified learning formalism, and the associated cross-environment insights goes beyond a simple "combination" of existing ideas, and provides a useful foundation for future work on general-purpose, environment-adaptive agents.
>
> ## Response to Q6–Q7: Extensions on modality and scenarios
> Finally, we address the questions about extending AUTOENV beyond text-only, and towards non-stationary or competitive scenarios.
>
> ### Multimodal and embodied extensions (Q6).
> In the revised manuscript, we add multimodal SkinEnv generation experiments (Appendix E.4), where the same underlying BaseEnv and ObsEnv are paired with different SkinEnv realizations, including text-only and multimodal variants. This demonstrates how the three-layer abstraction naturally supports richer observation modalities:
>
> - BaseEnv encodes the underlying world dynamics;
>
> - ObsEnv decides which aspects of the state are exposed;
>
> - SkinEnv handles rendering (text, diagrams, images) and natural-language instructions.
>
> To extend AUTOENV to 3D embodied or visual environments, one would replace BaseEnv with wrappers around a 3D or physics simulator, and design ObsEnv / SkinEnv to expose visual embeddings or pixel observations plus accompanying textual hints. We outline this path in the discussion as a concrete future direction.
>
> ### Non-stationary and competitive environments (Q7).
> The same abstraction can be used to define multi-agent or competitive settings by specifying in the theme that multiple agents interact under shared rules. For example, environments like 35_BattlefieldTactics can be instantiated as two-agent games where each agent uses its own policy while sharing the same BaseEnv dynamics. Non-stationarities (e.g., changing opponent strategies or evolving rules) can be modeled by letting BaseEnv dynamics depend on time or opponent behavior, while ObsEnv and SkinEnv control what each agent perceives.
>
> While our current experiments focus on single-agent LLM settings, we believe these extensions are straightforward within AUTOENV’s design. We have updated the discussion to make this clearer, and we are excited about exploring such scenarios in future work.

---

### Official Review · Reviewer_RJQ1 · 2025-10-28

**Soundness:** 2
**Presentation:** 1
**Contribution:** 3
**Rating:** 4
**Confidence:** 4

**Summary:**

The paper addresses the limitation of human-provided environments by proposing an automated environment generation method to prompt LLMs to produce code for constructing environments with distinct rule distributions and build a dataset of 36 heterogeneous environments with 358 levels.

Then the paper addresses the limitation of static learning strategies in different environments by formalizing the agent learning process as composable components and introducing selection strategies, optimization signals, and target components for learning adaptation analysis.

Experiments across 7 language models and 8 learning strategies demonstrate the quality of generated environment datasets and highlight the necessity of environment-specific learning strategy selection (different environments correspond to different optimal learning strategy configuration).

**Strengths:**

1. The paper addresses two core problems in current LLM agentic learning: (1) insufficient diversity of existing agent environments, and (2)  lack of exploration of different learning strategies in different environments. Solving these problems are crucial to develop scalable and optimized LLM agents in dynamic interactive environments.
2.  The paper constructs a dataset (AUTOENV-36) comprising 36 heterogeneous environments with 358 levels with varied reward, observation, and state dynamics.
3. The paper proposes a structured framework to analyze what learning strategies can be combined optimally in different environments, establishing a foundation for methodical evaluation of different learning methods across diverse environments.
4. Experiments demonstrate the quality of generated environments compared to human-supervised datasets, the usefulness of AUTOENV-36 to evaluate LLMs in agent capabilities, and the necessity to dynamically adapt and combine learning strategies in different environments (however, the evaluation metrics used in experiments are not well explained, raising concerns about the the credibility of the experimental results. See weaknesses below.).

**Weaknesses:**

1. The major concern about this paper is that it **lacks many details** to help understand the proposed concepts, metrics, and implementaion details. Here are some issues:

(1) Lack of specific calculation formula of evaluation metrics, such as the average generation cost per environment (cost in table 4), and  the "execution cost of optimized candidates in USD" in line 328, as well as "error rates across generation phases including execution errors, validation errors, and consistency errors and the overall success rates". By the way, what is USD? These raise concerns about the credibility and authenticity of experimental results.

(2) Lack of explanation and calculation formula of "theoretical reward upper bound".

(3) For optimization units, lack of detailed explanation and examples of "agent implementations and models"? The paper mentions somewhere that the agent may be "agent code" or "agent memory", which is confusing.

(4) Lack of explanation of "inverse semantics".

(5) Lack of descriptions and examples of "self-repair tasks"?

(6) No descirption of how to do "human review of LLM-generated requirements" concerning table 2, and how to ensure the consistency and reliability of the human review.

(7) No description of how to do "comprehensive feature analysis" to select environments to build AUTOENV-36 dataset. What features are analyzed and why chooses these features.

(8) Please provide specific temparature values set to different models in section 5.1.

2. The second major concern is **the practicality of generated environments.**
As shown in Table 1, the generated environments have an average of 6.10 available actions, which is a small action space compared to many realistic agentic tasks (e.g, web search/deep research/embodied interaction), especially open-ended environments with infinite action spaces (common in text-based environments such as dialogue).
And it is better to provide detailed description for each of the 36 generated environments (such as its state/action space and reward functions) and visualized examples for some of the environments to validate their practicality.
3. Another concern is the **representativeness of the 8 optimization methods** evaluated in experiments.
It seems that these optimization methods all belong to prompt engineering, however, post-training approaches such as SFT and RL have been widely used for agent learning in dynamic environments. What if these training approaches are incorporated into the learning adaptation analysis?
3. Since claude-4-sonnet is used as the optimization model in table 4, it should be compared as one of the baselines.
4. Lack of agent structures (code) before optimization in environments in the appendix.
5. Desciptions or reference of some figures or tables are missing in the main paper, such as Figure 1 and Table 4.

**Questions:**

1. How to calculate the results of "overall" in Table 2? It seems that it is not an average of automated and supervised.
2. Why do inverse semantic environments often yield higher scores than aligned semantic environments across most models?
3. Does Oracle Learning Selection in table 3 refer to combining Dynamics + Prompt and Instruction + Prompt methods?

---

> ### Author Response · Authors · 2025-11-27
> **Response to Reviewer RJQ1 (1/3)**
>
> We thank the reviewer for the very careful and detailed feedback. We are glad that you found the problem formulation and overall framework valuable, and we fully agree that the initial version lacked important details on metrics, implementation, and learning components. In the revised manuscript, we have (i) added new experiments (including SFT-based learning and semantic inversion), (ii) clarified all evaluation metrics, cost calculations, and temperature settings, and (iii) expanded the description of the environment-generation pipeline, semantics, and the construction of AUTOENV-36. We address your comments point by point below.
>
> ## Response to W1(3)/W3/W4/W5/Q3: Learning components, optimization methods, and baselines
>
> ### Optimization units: agent implementations and models (W1(3), W5).
> In the revised paper, Section 4.1 formalizes the space of learnable components as including memory, agent code, prompts, and models. In our current experiments, we instantiate this space using only prompt and agent-code optimization, because these are the most practically accessible levers for today’s LLM agents.
>
> - We provide concrete examples of these optimizations in Appendix D.4 and D.5, where we show how prompts and agent implementations are rewritten over iterations in specific environments.
> - Because "agent code" is allowed to modify the full Python implementation of the agent, some agent-code updates naturally change how the agent stores and uses context (e.g., adding a longer or better-structured interaction history). These can be viewed as simple forms of memory modification inside the same component.
> - For completeness, we also show the initial prompts and agent code before optimization in Appendix C.3, so that readers can clearly see what is being changed during learning.
>
> This is why, in the original text, we used the phrase "agent implementations and models": in the formulation we allow all four types of components in principle, while in our current experiments we concretely operate on prompts and agent code (with some implicit memory changes inside the code).
>
> ### Representativeness of optimization methods & inclusion of SFT (W3).
>
> Our main focus is indeed on training-free agentic learning (optimizing prompts and agent code), but we fully agree that training-based methods are important. In the revision we therefore add a supervised fine-tuning (SFT) baseline:
>
> - In Table 4 (Section 5.3), we now evaluate five learning methods on a 6-environment subset:
>   - four training-free methods (combinations of dynamics/instruction × prompt/agent-code optimization), and one SFT method based on Qwen-2.5-7B.
>   - We also report an UpperBound that, for each environment, selects the best of these five methods.
> - Appendix E.2 describes the SFT setup in detail: how we collect training data from AUTOENV levels, how many trajectories we use, and the fine-tuning configuration.
>
> Empirically, SFT becomes the best single method on most of the 6 environments, and on average it improves over the IO baseline by about 8 points in normalized accuracy. However, some training-free methods remain competitive and are better on specific environments. The UpperBound (oracle that picks the best method per environment) is consistently higher than any single method, showing that no single learning strategy works best everywhere, and that the method–environment interaction is exactly what our framework is designed to measure.
>
> Finally, to address the concern about representativeness: in Appendix D.3, we also show how four classic agent-learning methods (SPO, GEPA, AFlow, DGM) can be expressed within our component-centric formulation. This illustrates that our framework is general enough to cover a broad range of existing approaches, including both training-free and training-based ones.
>
> ### Claude-4-Sonnet as a baseline (W4).
> We agree that Claude-4-Sonnet should also appear as a baseline in the evaluation tables. In the revised experiments, Claude-4-Sonnet is explicitly included as a baseline execution model in the environment evaluation table (Table 4).
>
> ### Learning Upper Bound & “UpperBound” in Table 4 (Q3).
> In Section 4.2, we now formally define the Learning Upper Bound as:
>
> > "the best performance achievable by any method in the current method set when we are allowed to choose a different method for each environment."
>
> Concretely, in Table 4 and the related experiments:
> - We consider a fixed set of learning methods (the four training-free methods plus SFT in the 6-environment study, and the full eight-method space in the larger analysis).
> - For each environment, the UpperBound score is computed by taking the maximum normalized accuracy over all methods in this set.
>
> We use this Learning Upper Bound as a reference point to quantify the gap between an ideal environment-adaptive learner that can select the best method per environment and any fixed learning strategy.

---

> ### Author Response · Authors · 2025-11-27
> **Response to Reviewer RJQ1 (2/3)**
>
> ## Response to Q2: Semantic inverse experiments
>
> We agree that Table 3 looks surprising: in AUTOENV-36, environments with inverse semantics sometimes have higher scores than those with aligned semantics. To understand this better, we added a semantic inversion experiment, described at the end of Section 5.2 and in Appendix E.3.
>
> - In this experiment, we take some aligned environments and apply a “Skin Inverse” change: we only invert the natural-language description (Skin), while keeping the rules, rewards, and level layouts exactly the same. This lets us isolate the effect of semantics alone.
> - The results (Appendix E.3, Table A10) show that after we invert the semantics, model performance drops by about 68.8% in relative terms. This means that inverse semantics do make the tasks harder when the underlying environment is fixed.
>
> Therefore, the higher scores of inverse environments in the original AUTOENV-36 are not because inverse semantics are easier. Instead, it is because those particular inverse environments happen to be structurally simpler (for example, fewer important state variables or more forgiving reward rules) than the aligned ones that remain in the benchmark. We now explain this explicitly in Section 5.2.
>
> ## Response to W1(1)(2)(8)/Q1: Clarifying evaluation metrics, cost calculations, and experimental settings
>
> We thank the reviewer for carefully pointing out that the initial version did not sufficiently explain our evaluation metrics and experimental setup. In the revised manuscript, we clarify these points in both the main text and the appendix:
>
> ### Verification metrics and "Overall" in Table 2 (W1(1) & Q1).
> In Section 3.2 (Lines 209–210), we now explicitly define the three verification stages used by AUTOENV to test whether an environment is valid, together with their associated metrics (execution success, level-generation success, and reliability via consistency checking). These definitions are aligned with the metrics used in the Environment Generation experiments in Section 5.2 and Table 2.
> For Table 2, we also clarify how the “Overall” row is computed. It is not a simple arithmetic average of the “Automated” and “Supervised” rows. Instead, it is a weighted aggregate over all 100 themes (75 automated + 25 supervised). For example, the overall success rate of 65% corresponds to (75×0.6+25×0.8)/100.
>
> We further clarify that cost refers to the monetary cost of calling closed-source model APIs during generation or execution, and that USD simply denotes US dollars.
>
> ### Reward upper bound (W1(2)).
> We agree that the term "theoretical reward upper bound" was misleading. In the revision, we consistently use the term “validator-estimated upper bound.” As described in Section 3.2 (Lines 207–208), the validator is an important component in AUTOENV's generation process: it reads the environment code and implements a solver that both
> (i) checks the correctness of rewards on validation levels, and
> (ii) estimates an upper bound on the maximum achievable reward for each level using a fixed algorithm (e.g., exhaustive search in small state spaces, or structured search/heuristics in larger ones).
>
> Because each validator is tailored to the specific environment and may only approximate the optimal strategy, we cannot guarantee that this bound is truly "theoretical"; we therefore explicitly treat it as a validator-estimated upper bound. In Appendix C.2, we add a concrete example explaining how a validator computes this upper bound for a particular environment, to make the concept more intuitive.
>
> ### Temperature and experimental settings (W1(8)).
>
> In the revised version, we also clarify our temperature settings. For all models, we use the default temperature recommended by each provider and do not perform additional tuning. We will add this clarification to Section 5.1 so that readers can more easily reproduce our setup.

---

> ### Author Response · Authors · 2025-11-27
> **Response to Reviewer RJQ1 (3/3)**
>
> ## Response to W1(4)(5)(6)(7)/W2: Environment features, validation pipeline, and practicality of AUTOENV-36
>
> ### Aligned vs. inverse semantics (W1(4))
> We now define "aligned" and "inverse" semantics directly in Section 3.3 (Line 234). An environment’s semantic representation is aligned when the natural-language Skin matches the underlying rules and rewards (e.g., "poison decreases health," and in code, stepping on poison indeed yields a negative reward). It is inverse when the description is intentionally counterintuitive (e.g., "poison restores health while water decreases health"), so that agents cannot rely solely on surface wording and must learn the reward structure from interaction. This definition is used consistently throughout the paper and is further examined in our semantic-control experiments (see Response to Q2 above and Section 5.2).
>
> ### Self-repair tasks and generation pipeline (W1(5))
> We have expanded the description of the generation pipeline in Section 3.2 and in Figure 2 (Line 205). After an environment theme is instantiated into a DSL and then translated into a YAML specification, coding agents generate:
> - the three-layer environment classes
> - a level generator, a validator, and concise agent-facing documentation
>
> These artifacts then enter a self-repair loop: the system runs syntax and execution tests, collects error messages and validation failures, and iteratively edits the code until all tests pass or a repair budget is exhausted.
>
> ### Human review of LLM-generated requirements (W1(6))
>
> In Appendix B.7, we add a detailed description of the human review process used for a subset of themes. We first show an example of an automatically generated theme (Lines 1258–1265), where the description only specifies a high-level setting and rough gameplay. During human review, annotators ask the language model to expand and refine this theme along several concrete dimensions, including: action space, logic, reward and termination conditions.
> Reviewers also explicitly check for rule inconsistencies or contradictions. The resulting, human-reviewed theme is much closer to a formal environment-specification document. We provide before/after examples in Appendix B.7 (Lines 1267–1322) to show how human edits improve clarity and internal consistency.
>
> ### Selection of AUTOENV-36 (W1(7))
>
> After the full verification pipeline, 65 environments pass all checks. Among these, some share similar rule structures and test very similar competencies. To construct AUTOENV-36, we perform a two-stage selection:
>
> - First, we retain environments that are non-trivial for strong models, prioritizing those with lower validation scores under DeepSeek-V3.1 so that the benchmark remains challenging.
>
> - Second, we select a subset that is balanced across reward, observation, and semantic properties. This yields a set of 36 environments that are both validated and diverse in their rule distributions and difficulty.
>
> We also add a detailed capability taxonomy for AUTOENV-36 in Appendix A7, where each environment is annotated with its targeted capacities (planning depth, memory demands, exploration difficulty, etc.), so that readers can see a more fine-grained breakdown of what AUTOENV-36 measures.
>
> ### Practicality of AUTOENV-36 and the small action-space concern (W2)
> We appreciate the concern about the average of 6.10 available actions reported in Table 1. Here, it is important to clarify that this number refers to the number of action types, not the number of concrete action instances. Many realistic tasks (e.g., web search, embodied manipulation, dialogue) also have a small set of action types—such as search(query) and browse(url) for web agents, or pick(x, y, z) for embodied agents—while their effective action spaces are large or infinite due to continuous or high-dimensional parameters.
>
> The same pattern holds in AUTOENV. For example, in Env_1, the agent has a single “investment” action type, but the space of possible parameter combinations (investment amounts, targets, time steps) is unbounded. Thus, the benchmark is not limited to trivial discrete choices; rather, it exposes complex dynamics and long-term consequences through parameterized actions.
>
> Regarding environment descriptions, we do not list the full state and action space for all 36 environments in the main text because each implementation averages over 400 lines of code, and including all details would make the paper unwieldy. Instead:
> - we provide a detailed case study of one environment’s full implementation in Appendix B.
> - we provide full implementations of all environments in the supplementary material.
>
> For visualization, following your suggestion, we also demonstrate multi-modal Skins for several environments in Appendix E.4, where we render the same underlying rules with text-only vs. multi-modal observations. This helps illustrate what the environments look like and how the three-layer abstraction supports richer, more practical observation spaces.

---

### Official Review · Reviewer_KP3t · 2025-10-30

**Soundness:** 2
**Presentation:** 1
**Contribution:** 2
**Rating:** 4
**Confidence:** 3

**Summary:**

The paper introduces **AUTOENV**, a framework for (i) automated environment generation via a three-layer abstraction (BaseEnv/ObsEnv/SkinEnv), (ii) a formalization of agentic learning as a compositional loop, and (iii) the **AUTOENV-36** benchmark. The authors report low-cost environment generation, clear performance stratification across LLMs, and gains from environment-specific learning strategies.

**Strengths:**

1. **Well-motivated framework with practical utility**
   The three-layer environment abstraction combined with a domain-specific language (DSL) enables scalable, low-cost environment generation ($4.12 per environment, 90% success rate). This addresses a real bottleneck in agent development where manual environment creation is expensive and limited in diversity.

2. **Compelling empirical evidence for environment-specific learning**
   The core finding that Oracle Learning Selection (selecting the best learning method per environment) achieves 14% improvement with 2 methods and 32% with 8 methods—strongly demonstrates that no single learning strategy works universally. This challenges common assumptions in the field and is well-supported by experiments across 36 diverse environments.

**Weaknesses:**

1. **Unclear Baseline Definitions**:
The baseline used in the experiments is not clearly defined. The reported "14% improvement over baseline" and "32% improvement over baseline" are ambiguous without specifying the exact baseline configuration. Clarification is needed on whether the baseline refers to a fixed learning setup or the best performing existing model baseline.
2. **Unknown Capability Coverage of AUTOENV-36**:
AUTOENV-36 lacks a systematic capability taxonomy. It is unclear what specific cognitive skills each automatically generated environment tests (e.g., planning depth, spatial reasoning, memory requirements). How does AUTOENV-36's capability coverage compare to existing benchmarks?
3. **Insufficient Analysis of Component Improvement Mechanisms**:
Although Appendix E.2 provides qualitative examples of learned prompts and agent code, the analysis of the learning process remains insufficient. It is unclear what percentage of iterations modify Prompt vs. Agent Code, whether certain environment types favor specific components, and how these modifications correlate with performance gains.

**Questions:**

**Q1 (Baseline Definition):** Can you explicitly define the baseline corresponding to each reported improvement?

**Q2 (Capability Taxonomy):** Can you provide a mapping from environments to cognitive capabilities beyond the features in Table 1? How would you categorize the 36 environments along dimensions like planning depth, memory demands, or exploration difficulty? This would help readers understand what agent competencies the benchmark actually measures.

**Q3 (Component Analysis):** Can you provide detailed analysis of the learning process: (a) What percentage of learning iterations modify Prompt vs. Agent Code? (b) Is there correlation between environment features and effective components?

---

> ### Author Response · Authors · 2025-11-27
> **Response to Reviewer KP3t (1/2)**
>
> We thank the reviewer for the careful reading of our paper and for the thoughtful, concrete suggestions. We are glad that you found the framework well-motivated and practically useful, and in the revised version we have substantially expanded both the experiments and the analysis to address your main concerns. In particular, we (i) clarify the baseline and improvement definitions, (ii) provide a more explicit capability taxonomy for AUTOENV-36 and its relation to existing benchmarks, and (iii) add a detailed case study of learning trajectories to make the behavior of different components more transparent. We address these points in turn below.
>
> **Response to W1/Q1: Baseline and Improvements Definitions**
>
> We thank the reviewer for pointing out that our previous wording around "14% / 32% improvement over baseline" was ambiguous. In the revised version, we clarify both what the baseline is and how we report improvements:
> - We now explicitly define the baseline IO as a ReAct-style agent without any additional learning structure (Line 347).
> - In the original submission, the numbers "14%" (on 36 environments) and "32%" (on 6 environments) referred to the average relative improvement of the best single learning method over IO, computed across environments. We agree that this relative-percentage phrasing was easy to misinterpret.
> - In the new version, we no longer use standalone "14% / 32% improvement over baseline" statements. Instead, we directly report absolute normalized accuracies and their gain on IO. Concretely, we now highlight that
>   - on the 6-environment subset, the best learning method (SFT) improves over IO by about 8 points in normalized accuracy;
>   - on all 36 environments, the best learning method (Dynamics + Prompt Optimization) improves over IO by about 3 points (Line 103).
> - We have removed the old "14% / 32% improvement over baseline" wording from the abstract and main text, and replaced it with explicit references to IO and to absolute differences in normalized accuracy, which we hope fully resolves the confusion in W1 and Q1.
>
> **Response to W2/Q2: Capability coverage and environment taxonomy**
>
> We agree that the initial version did not sufficiently spell out what capabilities AUTOENV-36 actually measures. In the revised version, we address this by adding a detailed capability taxonomy and per-environment mapping (Lines 253–254; Appendix A7). Appendix A7 now provides a table that, for each of the 36 environments, lists its key properties together with the cognitive skills it is intended to probe, using the following :
> 1. Navigation / Spatial reasoning
> 2. Partial observability & memory
> 3. Counterintuitive / inverted semantics
> 4. Control, resource management, and multi-objective tradeoffs
> 5. Pattern recognition / symbolic regularities
> 6. Planning and long-horizon reasoning
> 7. Multi-agent / adversarial interaction
>
> In addition, Section 3.3 has been updated to give a clearer explanation of the basic feature dimensions reported in Table 1, such as reward type, observation type, semantic alignment, number of actions, and level count, so that readers can better interpret the structural properties of AUTOENV-36.
>
> It is also worth emphasizing that the main goal of this work is to show that environments can be constructed automatically, so that other researchers can use AUTOENV to build their own tailored environment sets. Nevertheless, despite being automatically generated, AUTOENV-36 already covers capability families that are closely related to those in classic benchmarks, such as:
> - puzzle-like state manipulation with tight constraints (reminiscent of Sokoban-style reasoning)[1],
> - grid-based navigation and hazard avoidance (as in FrozenLake-style environments)[2],
> - multi-step household or procedural tasks (inspired by ALFWorld-like scenarios) [3], and
> - multi-stage experiment or recipe execution (similar in spirit to ScienceWorld-style tasks) [4].
>
> [1] Sokoban: Enhancing general single-agent search methods: using domain knowledge
>
> [2] FronzenLake: The frozen lake problem. an example of optimization policy
>
> [3] ALFWorld: Aligning Text and Embodied Environments for Interactive Learning
>
> [4] ScienceWorld: Is your Agent Smarter than a 5th Grader?

---

> ### Author Response · Authors · 2025-11-27
> **Response to Reviewer KP3t (2/2)**
>
> **Response to W3/Q3: Component improvement mechanisms and learning analysis**
>
> We appreciate the reviewer’s request for a more detailed view of which components are actually modified during learning and how these modifications relate to performance gains. In the revised manuscript, we add a focused case study on Environment 19 (Backwards Valley Farm) in Appendix D.5.3, where we analyze learning trajectories under three optimization modes (instruction_prompt, instruction_agent, dynamics_agent):
> - What is modified.
>   - In the prompt-only configuration (instruction_prompt), all updates are constrained to the instruction prompt, while the ReAct agent code is frozen.
>   - In the agent-based configurations (instruction_agent, dynamics_agent), both the natural-language prompts and the associated agent implementation are updated, reflecting our definition of an “agent” component as code + tightly coupled prompts.
> - What patterns and effects we observe.
>   - For prompt-only learning, the instruction evolves from a generic description into a structured policy (e.g., explicit action priorities patterns), which alone yields substantial accuracy gains on this environment.
>   - For agent-level learning, updates systematically improve robustness (e.g., safer resets and logging) and introduce simple internal strategy structures (e.g., tracking past rewards and using “strategy insights” for future decisions), leading to further performance improvements at lower execution cost.
> This case study is intended to make the update patterns and their impact concrete, thereby clarifying how our component definitions translate into actual learning behavior.

---

### Official Review · Reviewer_GLTJ · 2025-11-01

**Soundness:** 2
**Presentation:** 1
**Contribution:** 1
**Rating:** 2
**Confidence:** 3

**Summary:**

The paper introduces a modular pipeline for automatically generating diverse RL-like environments and presents AUTOENV-36, a benchmark to evaluate agentic learning. The problem is timely and the engineering contribution is useful for the field, with clear motivation and reproducibility support. The work provides a promising benchmark direction, but clearer explanation of key concepts and more principled evaluation of learning strategies would strengthen its impact.

**Strengths:**

* The paper identifies a timely problem in agentic-learning - the lack of diverse, automatically generated environments and the need for adaptive learning strategies rather than fixed training pipelines.
* The authors provide detailed code snippets and algorithm descriptions that can help reproducibility.

**Weaknesses:**

While the motivation and system design are strong, I found the paper hard to follow. Several concepts are introduced without intuitive grounding or examples. For instance:
* Aligned vs. inverse semantics (Lines 223–229): These are referenced but not clearly explained; a concrete example would help clarify their role and importance.
* Selection strategies, optimization signals, and components (Line 269): Although briefly described later (Lines 299–309), the rationale behind the specific choices is not well justified. It is unclear what properties the chosen strategies are intended to probe, or why alternative agent-learning techniques were not explored.
* Execution model vs. optimization model (Lines 380–383): Their roles are not clearly defined; a short explanation of how these components interact would improve clarity.

* The paper also claims environment diversity but does not clearly describe how diversity is quantified (Line 371). Providing explicit metrics or qualitative analyses would strengthen the claim that AUTOENV-36 spans meaningfully different environment classes.

* A more detailed description of environments and model failure modes would make the paper easier to parse.

**Questions:**

Please see weaknesses.

---

> ### Author Response · Authors · 2025-11-27
> **Response to Reviewer GLTJ (1/2)**
>
> We sincerely thank the reviewer for the careful reading and for highlighting that, while the motivation and system design were strong, the initial version was hard to follow. We took this concern very seriously. In the revision, we have substantially **rewritten and reorganized the paper** to directly address the issues you raised:
>
> - We clarified the core concepts (aligned vs inverse semantics; selection strategies; optimization vs execution model) and added concrete examples and case studies.
> - We strengthened and formalized the learning formulation, explaining the rationale behind our choice of strategies and showing how existing methods fit into this framework.
> - We provided more quantitative analyses of environment diversity and added detailed descriptions of environments and model behaviors/failure modes.
>
> Below we respond to each of your points in turn.
>
> ### Aligned vs. inverse semantics
> > Aligned vs. inverse semantics (Lines 223–229): These are referenced but not clearly explained; a concrete example would help clarify their role and importance.
>
> In the revised manuscript, we have:
>
> - Added a clearer definition and example of aligned vs. inverse semantics in the environment section (Section 3.2–3.3). We now explain that
>   - aligned semantics describe rules in a way that matches human intuition (e.g., “harvesting ripe crops increases reward”), while
>   - inverse semantics describe the same underlying rules in a counter-intuitive way (e.g., “leave ripe crops untouched to gain reward, harvesting them is punished”).
>
> A concrete toy example is now given to illustrate how the same transition/reward dynamics can be presented with aligned or inverse textual descriptions.
>
> - Added a new semantic-control experiment on the Skin layer (“Skin inverse”) in Section 5.2 and Appendix E.3. There, we take environments with aligned semantics and only invert the Skin semantics, keeping BaseEnv dynamics and rewards fixed. We show that this inversion substantially increases task difficulty and also explain why inverse environments can have higher raw scores in the original AUTOENV-36 (they were generated to be structurally simpler).
>
> These changes both define the concept more intuitively and demonstrate empirically why aligned vs. inverse semantics matter.
>
> ### Selection strategies, optimization signals, and components
> > Selection strategies, optimization signals, and components (Line 269): Although briefly described later (Lines 299–309), the rationale behind the specific choices is not well justified. It is unclear what properties the chosen strategies are intended to probe, or why alternative agent-learning techniques were not explored.
>
> We agree that the original explanation was too high-level. In the revision:
> - We reorganized the learning formulation in Section 4 into a component-centric view with four basic objects (candidates, components, trajectories, metrics) and three stages (Selection, Optimization, Evaluation). For each stage, we now give intuitive explanations and examples (e.g., how trajectories and metrics are used in Selection vs. Optimization).
> - We clarify the rationale behind the specific selection strategies and optimization signals we chose:
>   - Some strategies are designed to probe simple, heuristic selection (e.g., greedy best-so-far).
>   - Others target pareto-style trade-offs between performance and cost, or stochastic exploration of candidate space.
>   - Optimization signals range from LLM-based self-reflection to explicit reward-based feedback, to highlight how different forms of feedback drive different learning dynamics.
> - Importantly, in **Appendix D.3**, we now show how four representative existing methods—**SPO, GEPA, AFlow, and Darwin Gödel Machine**—can be expressed within our framework as different combinations of:
>   - what is treated as the learnable component (prompts, workflow/agent code, etc.),
>   - how selection is performed (current best, pareto front, tree search, etc.), and
>   - what optimization signal is used (LLM reflection, benchmark scores, attribution to errors, etc.).
>
> This unification is intended to clarify that our framework is not an arbitrary list of strategies, but a structured way to position both our own variants and several existing agent-learning methods in a shared space. We also explicitly acknowledge that our experiments do not exhaust all possible agent-learning techniques, and we view AUTOENV as a testbed where additional methods (e.g., more advanced RL-style training) can be plugged into the same formulation.

---

> ### Author Response · Authors · 2025-11-27
> **Response to Reviewer GLTJ (2/2)**
>
> ### Execution model vs. optimization model
> > Execution model vs. optimization model (Lines 380–383): Their roles are not clearly defined; a short explanation of how these components interact would improve clarity.
>
> In the revised version, we now make the distinction explicit in Section 5.1 and Section 4:
> - The execution model is the model that acts inside the environment, i.e., it runs the ReAct-style policy (or its learned variants) and generates trajectories.
> - The optimization model is the model that observes candidates and trajectories and proposes updates to the learnable components (e.g., prompts or agent code) according to the selection/optimization procedure.
>
> We also add a short, concrete description of a typical loop:
> 1. he optimization model reads the current candidate(s), relevant trajectories, and metrics.
> 2. It proposes a modified candidate (e.g., refined instruction prompt or updated agent code).
> 3. The execution model runs this candidate in the environment to collect new trajectories and metrics.
> 4. These are fed back into the selection and optimization stages.
>
> In our main experiments, Claude-4-Sonnet is used as the optimization model, while other models (e.g., Gemini-2.5-Flash, DeepSeek-V3.1) serve as execution models. This separation is now explained explicitly where we introduce the models and in the description of the learning loop.
>
>
> ### Environment diversity and how it is quantified
> > The paper also claims environment diversity but does not clearly describe how diversity is quantified (Line 371). Providing explicit metrics or qualitative analyses would strengthen the claim that AUTOENV-36 spans meaningfully different environment classes.
>
> We agree that the first version did not make the diversity analysis explicit enough. In the revision, we:
>
> - Keep and clarify Table 1, which reports aggregate statistics across three dimensions—reward, observation, and semantics—as well as the number of actions, code size, and level counts.
> - Add a new capability and feature summary for each of the 36 environments in Appendix A7. For each environment, we indicate its reward structure, observation and partial observability properties, semantic alignment, and high-level capability tags (e.g., planning depth, memory demands, spatial reasoning, combinatorial/resource management).
>
> Together, these quantitative and qualitative analyses are meant to demonstrate that AUTOENV-36 is not just a collection of minor variants, but covers meaningfully different environment classes in terms of rules, dynamics, and learning difficulty.
>
> ### More detailed descriptions of environments and model failure modes
>
> We fully agree. To address this, we:
>
> - Added more concrete environment descriptions and examples in the main text (Section 3), including textual summaries and sketches of the state/action spaces for representative environments.
>
> - Expanded the appendices with environment-level details, including:
>
>   - example environment specifications and code structures (Appendix B.1–B.3),
>
>   - example SkinEnv renderings (including multi-modal variants) in Appendix A1, and
>
>   - a more detailed breakdown of environment generation success and failure modes in Appendix B.7 and E.3.
>
> - Included case studies of model behavior and failure modes in specific environments, showing how certain models struggle with, for example, counter-intuitive reward rules (inverse semantics), long-horizon dependencies, or partial observability.
>
> These additions are intended to make the system more concrete and to help readers build intuition about what the environments look like and why different models fail in different ways.
>
> -----
>
> In summary, your review correctly identified that the original submission’s main weakness was expository clarity, not motivation or system design. We have therefore invested significant effort into clarifying the key concepts, formalizing the learning framework, quantifying environment diversity, and adding concrete examples of environments and learning behaviors. We hope that these substantial revisions address your concerns and make the methodological and empirical contributions of AUTOENV much clearer.

---

### Author Response · Authors · 2025-11-27
**Public Comment**

We would like to sincerely thank the reviewers for their thoughtful and detailed feedback. We learned a great deal from the reviews and, building on the original core structure, we have substantially rewritten the manuscript. The revised paper now more clearly positions AUTOENV (i) as a platform for automatically constructing diverse agent environments, and (ii) as a unified formulation and testbed for existing agent learning methods. By scaling up both the environment set and the learning experiments, we aim to draw sharper insights about cross-environment agent learning and the gap between fixed learners and environment-adaptive strategies.

Below we briefly summarize the main experimental and presentation changes.

**For experiments and insights**
- Learning experiments
  -  (For Reviewer RJQ1, FGMv.) We added a new 6-environment learning study using an open-source model, where we introduce SFT into our learning experiments and compare it with four training-free methods (Table 4, Section 5.3; SFT setup in Appendix E.2). This shows how training-based approaches behave within the same component-centric formulation.
  -  (For Reviewer RJQ1, KP3t.) We then scale the learning experiments to all 36 environments, using Gemini-2.5-Flash to measure how performance changes with environment diversity and method diversity (Figure 3 and Table A9, Section 5.3). We study both the gains of individual methods and the effect of expanding the method set on the Learning Upper Bound (an oracle that picks the best method per environment).
  - Together, these experiments support three key insights:
    - different environments favor different learning methods (training is not uniformly better than training-free);
    - the benefit of any fixed learning method decays as the number and diversity of environments increase;
    - simple environment-adaptive selection across methods substantially improves learning performance, yet still leaves a clear gap to effective cross-environment learning.

- Environment generation experiments
  - (For Reviewer RJQ1, GLTJ.) We added a semantic-control experiment on the Skin layer (“Skin inverse”), where we invert only the Skin semantics while keeping BaseEnv dynamics and rewards fixed. This demonstrates that inverse semantics substantially increase task difficulty (Section 5.2; detailed ablation in Appendix E.3).
  - (For Reviewer FGMv.) We further present multi-modal SkinEnv generation examples, combining the same underlying rules with different observational skins (text-only vs. multi-modal). This illustrates that the three-layer abstraction allows us to decouple rules from observations and, under fixed dynamics, construct environments of different difficulty via Skin-level semantic inversion and multi-modal rendering (Section 3.3; examples in Appendix A1).

**For presentation**
- (For Reviewer FGMv.) We strengthened the framing of AUTOENV and AUTOENV-36 as a testbed for revealing the limitations of current cross-environment agent learning, and we explicitly use the new experiments to diagnose where existing learning methods fail. Based on this, we rewrote the abstract and introduction to focus on cross-environment agent learning.
- (For Reviewer GLTJ, KP3t.) We reorganized the method section to provide more concrete details of how AUTOENV constructs environments, and we added a new pipeline figure illustrating the automatic construction process together with representative generated environments (Figure 2, Section 3). We also added a more detailed description and distributional analysis of the features of the AUTOENV-36 dataset (Appendix A7).
- (For Reviewer KP3t.) In parallel, we clarified the learning formulation and, in the appendix, we show how four classic agent learning methods (e.g., SPO, GEPA, AFlow, DGM) can be expressed within this formulation (Section 4; Appendix D.3).
- (For Reviewer KP3t.) We aligned and surfaced more experimental details in the main text, including clearer metric definitions and experimental settings, and we added direct references to the new Table 4, Figure 3, and several appendices that provide further implementation and training details (Sections 5.1–5.3; Appendix C–E).

In addition, we provide point-by-point responses under each individual review, where we aim to address specific questions and concerns with direct references to the revised sections and appendices.

---

### Meta-Review · Area_Chair_Rhw3 · 2025-12-08

**Summary:**

Clarity & definitions. Paper was hard to follow: unclear aligned vs. inverse semantics, selection strategies/optimization signals/components, and the roles of execution vs. optimization models.

Baselines & metrics. Ambiguous % improvement over baseline; unclear IO baseline, temperature settings

Diversity & capability coverage. AUTOENV-36’s diversity and what capabilities it measures weren’t quantified

Learning mechanisms. Insufficient analysis of what is actually modified, and how those changes relate to gains

Presentation gaps. Missing/weak figure references, limited environment descriptions and failure modes; request to include Claude-4-Sonnet as a baseline.

Contribution framing. One reviewer viewed the work as incremental (combination of known pieces) rather than a fundamental advance.

**Reviewer Concerns:**

Clarity of concepts (aligned/inverse, strategies, exec vs. opt model): Addressed

Baseline & metrics (IO, % gains, LUB, costs, temps, “Overall”): Addressed (replacement of vague % claims by absolute normalized accuracies vs IO)

Diversity & capability coverage: Addressed

Learning mechanisms (what changes, why it helps): Only partially addressed (still no aggregate stats across all envs on % prompt vs code edits or correlations with capability tags)

Broader learning baselines (SFT/RL): Partially addressed (SFT not uniformly best. No RL baselines)
Practicality & robustness: Addressed

Presentation gaps (fig refs, env descriptions, failure modes, Claude baseline): Addressed

Contribution framing (incremental vs. substantive): Partially addressed (no new theory, but stronger unification and scale)

**Reviewer Scores:**

I cannot speculate how reviewers would have revised the scores, especially as I cannot see their reactions to the authors' responses. But all four reviewers started from a reject position...

---

### Decision · Program_Chairs · 2026-01-26

Reject